# Development of long lifespan high-energy aqueous organic‖iodine rechargeable batteries

Zishuai Zhang [1,2,3], Yilong Zhu[4], Miao Yu [1] ✉, Yan Jiao [4] ✉ & Yan Huang [2,3,5] ✉

Rechargeable aqueous metal‖$I_2$ electrochemical energy storage systems are a cost-effective alternative to conventional transition-metal-based batteries for grid energy storage. However, the growth of unfavorable metallic deposition and the irreversible formation of electrochemically inactive by-products at the negative electrode during cycling hinder their development. To circumvent these drawbacks, herein we propose 3,4,9,10-perylenetetracarboxylic diimide (PTCDI) as negative electrode active material and a saturated mixed KCl/$I_2$ aqueous electrolyte solution. The use of these components allows for exploiting two sequential reversible electrochemical reactions in a single cell. Indeed, when they are tested in combination with an active carbon-enveloped $I_2$ electrode in a glass cell configuration, we report an initial specific discharge capacity of 900 mAh g$^{-1}$ (electrode mass of iodine only) and an average cell discharge voltage of 1.25 V at 40 A g$^{-1}$ and 25 ± 1 °C. Finally, we also report the assembly and testing of a PTCDI|KCl-$I_2$|carbon paper multilayer pouch cell prototype with a discharge capacity retention of about 70% after 900 cycles at 80 mA and 25 ± 1 °C.

In recent years, rechargeable metal-halogen batteries, which rely on strict redox chemistry to achieve high energy and power density, have attracted considerable attention[1-6]. In particular, solid iodine ($I_2$) has shown better operability and (electro)chemical stability than its liquid bromine and gaseous chlorine analogs[7-10]. Because of its high abundance[4], high theoretical capacity (422 mAh g$^{-1}$), and theoretical redox potential of $I^0/I^+$ (1.07 V vs. standard hydrogen electrode), $I_2$ stands out as a particularly promising cathode material for aqueous batteries[3-6,11-15]. Moreover, the solid–liquid $I^-/I^0/I^+$ conversion of the $I_2$ cathode avoids the issue of electrode structural degradation that commonly occurs in other intercalation materials[5,6,14]. Major success has been achieved in the development of various aqueous metal‖$I_2$ batteries, e.g., Fe‖$I_2$[16], Al‖$I_2$[17] and Zn‖$I_2$[3,5,6,18–24]. However, these batteries

are affected by the detrimental (electro)chemical behavior of the metal electrode upon cycling[3–6,12–21,25]. The dendrite growth and corrosion of the metal anodes cause rapid attenuation of the capacity and short circuit; the iodine anionic species lead to the formation of electro-chemically inactive complexes with the metal anode, which induces the irreversibility of the $I_2$ cathode and limited lifespan, similar to the shuttle effect in lithium-sulfur batteries[3,4,26].

To address these issues, considerable efforts have been made to modify the metal anode and firmly anchor iodine species onto the cathode[5,6,12,14,16,27,28], leading to significant improvements. The pre-viously reported results collected from the developed aqueous metal‖$I_2$ batteries are 35,000 cycles at 10.55 A g$^{-1}$ (G-Zn‖ZC-mK$_2$CO$_3$@$I_2$)[21], 23,000 cycles at 6 A g$^{-1}$ (Zn‖$I_2$-Nb$_2$CT$_x$)[6], and 6000 cycles at 1.92 A g$^{-1}$

$^1$State Key Laboratory of Urban Water Resource and Environment, School of Chemistry and Chemical Engineering, Harbin Institute of Technology, Harbin 150001, China. $^2$Sauvage Laboratory for Smart Materials, School of Materials Science and Engineering, Harbin Institute of Technology, Shenzhen 518055, China. $^3$Shenzhen Key Laboratory of Flexible Printed Electronics Technology, Harbin Institute of Technology, Shenzhen 518055, China. $^4$School of Chemical Engineering and Advanced Materials, The University of Adelaide, Adelaide, SA 5005, Australia. $^5$State Key Laboratory of Advanced Welding and Joining, Harbin Institute of Technology, Harbin 150001, China. ✉e-mail: miaoyu_che@hit.edu.cn; yan.jiao@adelaide.edu.au; yanhuanglib@hit.edu.cn

(Zn-BTC||I$_2$)[18] for cycling; 109,100 W kg$^{-1}$ at 179 Wh kg$^{-1}$ (Zn||Co[Co$_{1/4}$Fe$_{3/4}$(CN)$_6$]/I$_2$)[20] and 410 Wh kg$^{-1}$ at 110 W kg$^{-1}$ (G-Zn||ZC-mK$_2$CO$_3$@I$_2$)[21] for the specific energy and power performance; and 140 mAh g$^{-1}$ at 21.1 A g$^{-1}$ (G-Zn||ZC-mK$_2$CO$_3$@I$_2$)[21] and 419 mAh g$^{-1}$ at 2 A g$^{-1}$ (Zn||PAC-I$_2$)[3] for the high specific current performance. However, it remains challenging to substantially increase the performance of metal||I$_2$ batteries. A novel alternative system has been proposed recently, i.e., replacing the anode materials with hydrogen (H$_2$)[29]. The H$_2$||I$_2$ system shows promise but has certain limits: It is limited to Swagelok cell type for operation and involve time-consuming fabrication process also using costly Pt-based catalysts. Furthermore, the cell discharge voltage is <1.2 V, and the capacity is ~0.5 mAh at 2.5 mA cm$^{-2}$. A breakthrough strategy is, therefore, needed to improve the lifespan, energy and power content of I$_2$-based batteries.

Connection in parallel of batteries is a common route to attain higher capacity and energy. However, when batteries are connected through external wires, their performance is often compromised[30]. In this regard, integrating more than one reversible redox reaction in a single electrochemical energy storage cell is an advantageous strategy to avoid parallel connection of multiple cells (with single redox reaction) to improve the capacity outcome of the battery system. Recently, Dai et al. proposed a "cascade" battery (i.e., a single cell bringing together at least two sequential reversible electrochemical reactions) based on a Zn||S system[30]. Nevertheless, the feasibility of extending the cascade concept to other electrochemical energy storage is not implemented yet.

Herein, we report an aqueous organic||I$_2$ battery with cascade concept. Distinct from the previously reported aqueous metal||I$_2$ and H$_2$||I$_2$ battery systems, 3,4,9,10−perylenetetracarboxylic diimide (PTCDI) is employed as the active material at the negative electrode. Because PTCDI is inert to various iodine anionic species and the fast

conversion of I$^-$/I$^0$/I$^+$ in the saturated potassium chloride (KCl) electrolyte, a long lifespan (92,000 cycles at 40 A g$^{-1}$) and appealing discharge capacity performance at high currents (e.g., 104 mAh g$^{-1}$ at 160 A g$^{-1}$), high specific energy (434 Wh kg$^{-1}$ at 40 A g$^{-1}$) and power (155,072 W kg$^{-1}$ at 160 A g$^{-1}$) can be delivered. Moreover, by using a saturated mixed of KCl/I$_2$ aqueous solution as the electrolyte, the high cut-off voltage further reaches 2.5 V.

## Results

The development of an efficient I$_2$-cathode battery depends on two factors: the electrolyte and anode. The selection of an aqueous electrolyte should guarantee the highest conversion efficiency of I$^-$/I$^0$/I$^+$ to fully exploit the high theoretical capacity and redox potential of the I$_2$ cathode. That is, for the I$^-$/I$^0$ conversion, the cation in the electrolyte must reduce the conversion energy barrier. As a result, the discharge cell voltage could be promoted due to the reduced polarization[5]. For the I$^0$/I$^+$ conversion, the anion in the selected electrolyte must facilitate the dissociation of I$^+$ compounds to accelerate the I$^0$/I$^+$ conversion[2,3,5]. Regarding the anode, only when it simultaneously satisfies the demands of structural stability, inertness to various iodine anionic species, low redox potential, high capacity and low cost can it guarantee a long lifespan with an improved energy outcome. Compared with inorganic materials, organic materials featuring structural diversity, flexible molecular structure and low cost are promising candidates for aqueous I$_2$-cathode battery systems[31–33].

Density functional theory (DFT) calculations were performed to predict the reaction energy profiles of I$^-$/I$^0$ conversion in the presence of different cations within the electrolyte for the selection of cation candidates. In addition, the dissociation energy of I$_n$X (where X denotes the anion of the electrolyte and n denotes the electron number) was compared in different anion environments to identify the best candidate. As shown in Fig. 1a, the lowest value of the Gibbs free energy change ($\Delta G$) was observed when the I$^-$/I$^0$ conversion occurred in a K-ion environment. Thus, the aqueous electrolyte containing K$^+$ should be considered an ideal candidate for reversible redox reactions with fast reaction rate of the I$^-$/I$^0$ conversion[5,20]. For anions, the lowest dissociation energy of ICl was observed when compared to that of other iodine compounds, indicating the fast conversion of I$^0$/I$^+$ in aqueous electrolyte containing Cl$^-$ (Fig. 1b)[2,3,5]. Accordingly, the aqueous KCl electrolyte should guarantee the highly reversible redox reaction of the I$_2$ cathode. Based on Fig. 1a, b, iodine-containing reactions can be performed entirely in the aqueous KCl + I$_2$ mixed electrolyte. Considering the different reaction potentials between I$^-$/I$^0$ (0.53 V vs. SHE) and I$^0$/I$^+$ (1.07 V vs. SHE)[3], a I$^-$/I$^0$||I$^0$/I$^+$ redox system should be self-constructed in the mixed KCl/I$_2$ aqueous electrolyte. Moreover, the linear sweep voltammetry (LSV) test showed that the total operational voltage window was up to 2.58 V in a saturated KCl aqueous electrolyte (3.4 mol l$^{-1}$) (Supplementary Fig. 1)[12,34,35].

Then, it was necessary to determine an appropriate anode that can efficiently store and release K ions in an aqueous electrolyte. Due to the sluggish kinetics of the larger Cl$^-$ compared with K$^+$, materials capable of intercalation/deintercalation of Cl$^-$ were ruled out[36]. As shown in Fig. 1c, organic aromatic molecules with various numbers of aromatic rings and carbonyl groups (Dipyridophenazine (DPPZ)[37], β-perylene-3,4,9,10-tetracarboxylic dianhydride (β-PTCDA)[37], 1,4,5,8-naphthalene-tetracarboxylic dianhydride-derived polyimide (PNTCDA)[38], 5,7,12,14-pentacenetetrone (PT)[33] and inorganic and composite materials (KTi$_2$(PO$_4$)$_3$[39], potassium Prussian blue (KPB)@ polypyrrole (PPy)[40]) were compared. In this regard, PTCDI displayed the lowest redox potential and highest specific capacity, thus it can be considered a viable negative electrode active material for aqueous battery system with KCl-based electrolytes[34,41]. X-ray diffraction (XRD) and Fourier transform infrared (FT-IR) spectroscopy were performed to determine the crystal structure and functional groups of PTCDI (Supplementary Fig. 2, Supplementary Note 1 and Supplementary Table 1). In addition,

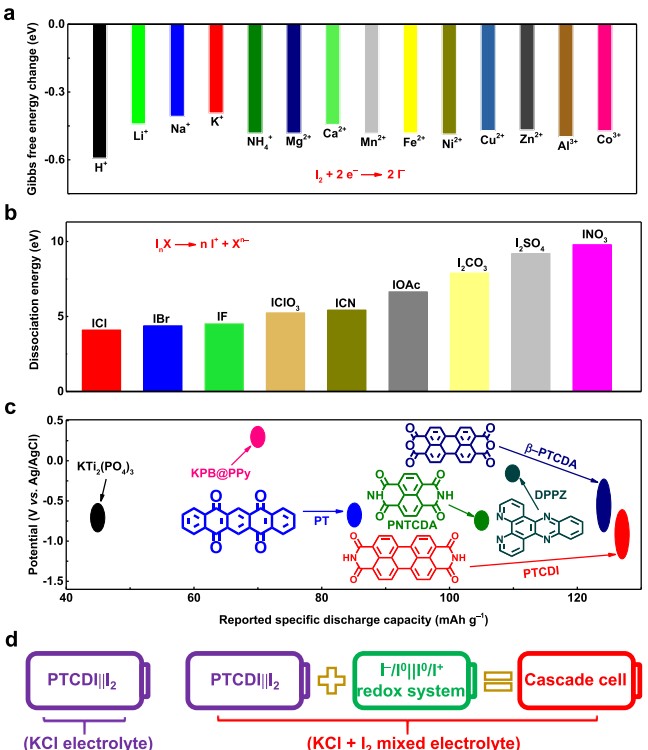

**Fig. 1 | Theoretical calculation of selected species and comparison of selected molecules and schematics of cascade cell. a** Comparison of the calculated Gibbs free energy change ($\Delta G$) of I$^-$/I$^0$ conversion in different cation environments. **b** Comparison of the calculated dissociation energy of various iodine compounds. **c** The charge/discharge potential range and reported specific discharge capacity of anodes with K-ion insertion/deinsertion chemistry. **d** A schematic of the PTCDI||I$_2$ single cell and a cascade cell.

the electrochemical performance of the PTCDI electrode was investigated in a three-electrode glass cell configuration (counter electrode: platinum foil; reference electrode: standard Ag/AgCl; electrolyte: saturated KCl solution) at $25 \pm 1\,°C$ (Supplementary Figs. 3 and 4 and Supplementary Note 2 and 3). These results confirmed the reversible insertion/extraction of K-ion owing to the interplanar distance of PTCDI (Supplementary Fig. 5 and Supplementary Note 4)[34,42,43]. Moreover, the CV curves of the PTCDI electrode in a saturated mixed $KCl/I_2$ aqueous electrolyte were almost identical to that in saturated KCl aqueous electrolyte and the series resistance (obtained by the analysis of the electrochemical impedance spectroscopy (EIS) measurements) remained almost the same, verifying that the PTCDI electrode was inert to various iodine anionic species (Supplementary Fig. 6, Supplementary Table 2 and Supplementary Note 5). Active carbon-enveloped $I_2$ ($I_2$@AC) (the content of $I_2$ was 47.2%) synthesized through a facile physical adsorption method was utilized as the active material at the positive electrode (Supplementary Fig. 7 and Supplementary Note 6). When the PTCDI electrode and the $I_2$@AC electrode are coupled as a full cell in a two-electrode glass cell configuration, the full cell can not only work stably in a saturated KCl aqueous electrolyte as the PTCDI‖$I_2$ single cell (Eqs. (1)–(2)), but also work stably as a cascade cell in a saturated mixed $KCl/I_2$ aqueous electrolyte. The cascade cell consists of a PTCDI‖$I_2$ single cell chemistry (Eqs. (1)–(2)) and an $I^-/I^0‖I^0/I^+$ redox chemistry (Eqs. (3)–(4)) (Fig. 1d).

The PTCDI‖$I_2$ single glass cell in a saturated KCl aqueous electrolyte:

$$\text{Positive electrode}: 2\,I^- \leftrightarrow I_2 + 2\,e^-\ I_2 + 2\,Cl^- \leftrightarrow 2\,ICl + 2\,e^- \quad (1)$$

$$\text{Negative electrode}: 2\,PTCDI + 4\,K^+ + 4\,e^- \leftrightarrow 2\,PTCDI - 2K \quad (2)$$

The PTCDI‖$I_2$ cascade glass cell in a saturated mixed $KCl/I_2$ aqueous electrolyte:

Step 1: $I^-/I^0‖I^0/I^+$ redox chemistry:

$$\text{Positive electrode}: I_2 + 2\,Cl^- \leftrightarrow 2\,ICl + 2\,e^- \quad (3)$$

$$\text{Negative electrode}: I_2 + 2\,e^- \leftrightarrow 2\,I^- \quad (4)$$

Step 2: PTCDI‖$I_2$ single cell chemistry, reaction equations are the same as Eqs. (1)–(2).

Regarding the saturated KCl aqueous electrolyte, the aqueous PTCDI‖$I_2$ glass cell was assembled with PTCDI as negative electrode active material (the mass content of PTCDI in the electrode was 70% and the average mass loading of the electrode was $1.0-1.2\,mg\,cm^{-2}$) and $I_2$@AC as positive electrode active material (the mass content of $I_2$ in the electrode was 40% and the average mass loading of the electrode was $1.0-1.2\,mg\,cm^{-2}$). Considering the theoretical capacity of $I_2$ ($422\,mAh\,g^{-1}$)[3,9,28] and PTCDI ($137\,mAh\,g^{-1}$)[42], a mass ratio of PTCDI:$I_2$ = 4 was utilized. During the charge/discharge process, the oxidation/reduction reaction of $I^-/I^0/I^+$ occurred at the $I_2$@AC electrode, and the enolization/recovery of carbonyl groups occurred at the PTCDI electrode with $K^+$ intercalation/deintercalation (Supplementary Fig. 8a, b)[2,3,5]. Hence, the chemical equations of the PTCDI‖$I_2$ glass cell in a saturated KCl aqueous electrolyte can be described as Eqs. (1)–(2).

With an increase in the cycle number, the initial redox peaks gradually shifted toward higher voltage values and finally stabilized due to the activation process. (Supplementary Fig. 9a, b) Noted that the intersection of the EIS curve with the horizontal axis represents the series impedance and the intersection position shifts toward the small value direction of the horizontal axis during cycling, verifying the reduced series resistance of the cell during cycling and demonstrating the activation process (Supplementary Fig. 9c, Supplementary Table 3

and Supplementary Note 7)[26,44]. Consequently, typical CV curves of PTCDI‖$I_2$ within the voltage range of 0–2.4 V are presented (Fig. 2a). Oxidation peaks at 1.54 V, 2.24 V and reduction peaks of 1.71 V, 0.91 V, 0.72 V were clearly observed. For the $I_2$@AC electrode, the redox pair of 2.24 V/1.71 V corresponds to the reversible reaction of $I^0/I^+$ while the remaining redox pairs correspond to the reversible transformation between $I^-$ and $I^0$[2,3,5]. For the PTCDI electrode, the anodic and cathodic peaks correspond to the stepwise intercalation and deintercalation of $K^+$[34,41,43]. Additionally, the dominating capacitive charge storage behavior at high currents testify the high-power behavior of the PTCDI‖$I_2$ full cell (Supplementary Fig. 9d–g and Supplementary Note 7)[45,46]. The typical galvanostatic charge/discharge (GCD) curve obtained from the 1000th cycle displayed the discharge voltage plateau at 1.90 V and the calculated average discharge voltage of 1.26 V during the discharge process, and a discharge capacity of $324\,mAh\,g^{-1}$ with the coulombic efficiency of 69% was delivered at $40\,A\,g^{-1}$ (Fig. 2b). When the cut-off voltage of the cell was set to 0.3 V, a discharge capacity of $300\,mAh\,g^{-1}$ with the coulombic efficiency of 82% was delivered (Supplementary Fig. 10 and Supplementary Note 8). Hence, the cut-off voltage of the full cell had the effect on the specific capacity and coulombic efficiency to some extent. Considering that a slightly higher capacity output was obtained, the cut-off voltage of the full cell was set to 0 in this work. In addition, the maximum discharge voltage plateau of the PTCDI‖$I_2$ glass cell is higher than that reported for aqueous $I_2$-based cathode batteries and most aqueous rechargeable K-ion full batteries (ARKFBs) (Supplementary Fig. 11 and Supplementary Note 9).

As shown in Fig. 2c, the cycling performance of the PTCDI‖$I_2$ glass cell was further investigated in GCD mode at $40\,A\,g^{-1}$, and the specific capacity was calculated based on the mass of $I_2$ in the $I_2$@AC electrode. Due to the cell instability at specific currents $\leq 1\,A\,g^{-1}$ (Supplementary Fig. 12 and Supplementary Note 10), we decided to apply higher specific current to hinder the occurring of parasitic reactions kinetically[33,34,47,48]. Notably, ~810 $mAh\,g^{-1}$ of charge capacity was delivered in the first charge process and it decayed to around $400\,mAh\,g^{-1}$ in the following cycles. The initial charge capacity beyond the theoretical capacity was mainly derived from the irreversible reactions of impurity existing on the surface of electrode. Besides, there was a small increase of the discharge capacity in the initial cycles, which may attribute to the activation process. The activation process ascribes to the more adequate electrolyte infiltration and progressively stabilized structural mechanics during repeated charge/discharge process[44]. Moreover, a discharge capacity of $364\,mAh\,g^{-1}$ was delivered after the initial activation process. Such a high discharge capacity originated from the full use of $I^-/I^0/I^+$ conversion reactions[2,3,5]. A discharge capacity of $156\,mAh\,g^{-1}$ was achieved after 92,000 cycles at $40\,A\,g^{-1}$ (Fig. 2c). The discharge capacity retention from the 1st cycle to the last cycle (i.e., the 92,000th) was calculated to be 43%. The relatively low coulombic efficiency (CE) values and the decrease in the specific capacity with cycle number were mainly attributed to the solubility of iodine molecules and iodine anionic species derived from the $I_2$@AC cathode[3,5,6,13,14]. The PTCDI‖$I_2$ glass cell tested at high specific current demonstrate long-term performance which exceeds the state of the art of $I_2$-based aqueous batteries[3,5,6,16–21,49–55] and most ARKFBs[37,39–41] (Supplementary Table 4). Such a lifespan with high capacity at a high rate was attributed to the inertness to iodine anions and the layered structure of the PTCDI electrode with large interplanar spacing, as well as the stability of the PTCDI electrode at high oxidation potentials owing to the large π-conjugated structure[34,47,48]. In addition, the reversible enolization of quinone (−C=O) to quinone salts (−C−O−M) was propitious to maintain the structural stability of the PTCDI electrode during repeated cycles and thus favored the cycling performance of the full cell[34]. To determine the capacity contribution of active carbon (AC), GCD of the PTCDI‖AC full cell without iodine was tested under $40\,A\,g^{-1}$ at $25 \pm 1\,°C$, and a discharge capacity of only $38\,mAh\,g^{-1}$ was delivered, verifying the fact that the capacity

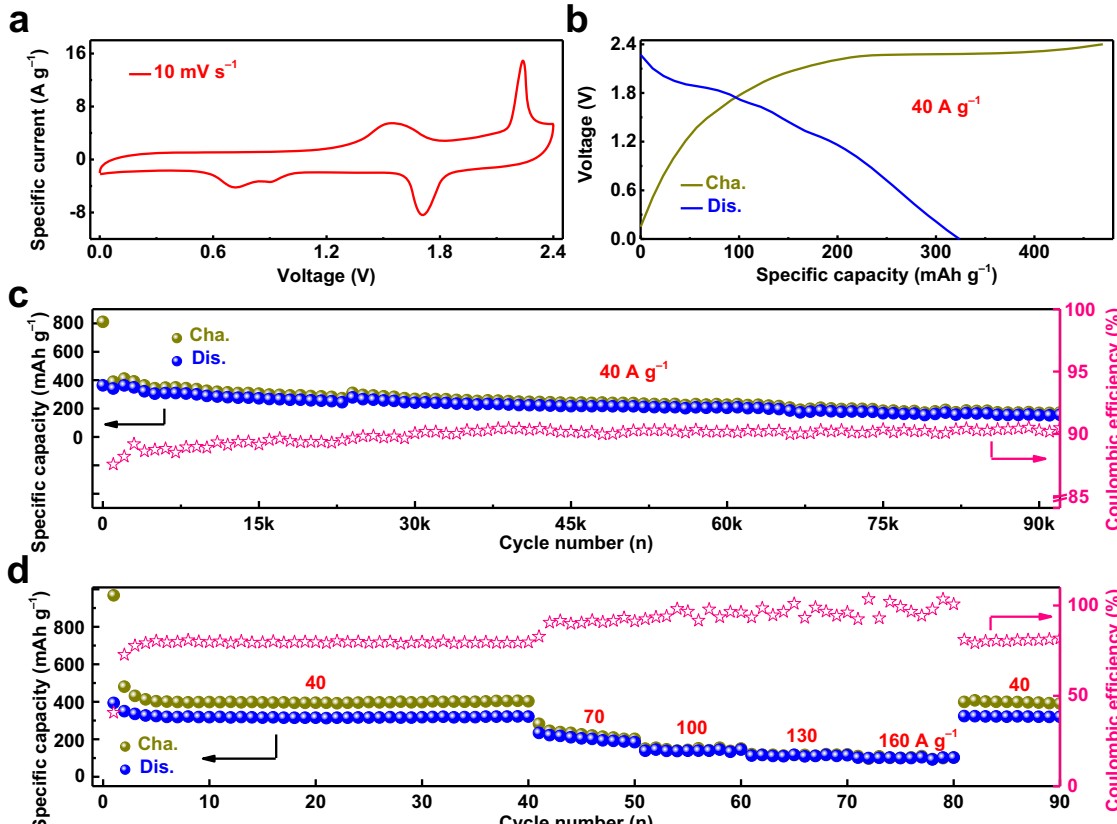

**Fig. 2 | Electrochemical performance of the PTCDI∥I₂ glass cell in a saturated KCl aqueous electrolyte. a, b** Typical CV and GCD profiles of the PTCDI∥I₂ glass cell, respectively. The specific capacity and specific current were calculated based on the mass loading of I₂ in the I₂@AC cathode. **c** Cycling performance of the full cell at 40 A g⁻¹. **d** Rate capability at different specific currents.

contribution of AC was not significant (Supplementary Fig. 13)[3]. Originated from the dissolubility of iodine and iodine species in the aqueous environment of the batteries, self-discharge behavior is common for the aqueous iodine-cathode battery systems[3–6]. How to reduce the self-discharge rate effectively has been an intriguing but challenging issue. In the cases of the conventional aqueous metal∥I₂ battery systems, iodine anionic species can diffuse into the vicinity of the metal anode and induce the formation of electrochemically inactive complexes, resulting in the irreversible loss of iodine elements and self-discharge behavior[3–6]. Encouragingly, the anode material adopted in this work was PTCDI, an organic compound featuring intrinsic inertness to various iodine anionic species. Iodine anionic species were unable to react with the PTCDI electrode to form electrochemically inactive complexes. As a result, the PTCDI∥I₂ glass cell in this work displayed a much lower self-discharge rate than that of the conventional Zn∥I₂ glass cell (Supplementary Fig. 14 and Supplementary Note 11). The self-discharge rate for the full cell was tested at the open circuit after fully charged. Specifically, the time required for a voltage drop of 0.4 V extended from 27.6 s for the Zn∥I₂ glass cell to 78.2 s for the PTCDI∥I₂ glass cell. That is, the self-discharge rate of the PTCDI∥I₂ glass cell was reduced to 35.3% as that of the Zn∥I₂ glass cell.

As illustrated in Fig. 2d, the rate capability of the PTCDI∥I₂ glass cell was investigated under various specific currents ranging from 40 A g⁻¹ to 160 A g⁻¹ at 25 ± 1 °C. In detail, discharge capacities of 323, 204, 140, 116 and 104 mAh g⁻¹ were delivered at 40, 70, 100, 130 and 160 A g⁻¹, respectively. A discharge capacity of 322 mAh g⁻¹ was achieved when the specific current was shifted from 160 A g⁻¹ to 40 A g⁻¹, which is 99.7% of the initial specific capacity, validating the rate capability of the full cell and agreeing with the CV results in Supplementary Fig. 9. The PTCDI∥I₂ glass cell tested at high specific current demonstrate improved performance in term of specific discharge

capacity when compared against I₂-based cathode batteries[3,5,6,16–21,49–55] and ARKFBs[37,39–41] (Supplementary Table 4). The full cell exhibited an improved specific energy of 434 Wh kg⁻¹ at a specific power of 50,420 W kg⁻¹ and an improved specific power of 155,072 W kg⁻¹ at a specific energy of 86 Wh kg⁻¹. These results are higher than those reported for aqueous batteries with I₂-based cathodes and most ARKFBs (Supplementary Fig. 15 and Supplementary Note 12). It is noted that the specific energy and power were both calculated based on the mass of the I₂ in the positive electrode. We are confident that our research work could be considered as an initial proof of concept for the development of high-energy aqueous halogen batteries with organic-based negative electrodes[33,56,57].

To identify the underlying redox chemistry of the two-electron transfer of the I₂ cathode, ex situ ultraviolet–visible (UV–vis) and ex situ Raman spectroscopy measurements were performed. Figure 3a displays the GCD curve collected from the 10th cycle of the PTCDI∥I₂ glass cell at 40 A g⁻¹ and 25 ± 1 °C with selected voltage points marked for characterization. As illustrated in Fig. 3b, a broad absorption peak in the range from 375 nm to 494 nm corresponded to the formation of iodine molecules when charged to 2.0 V, confirming the conversion from iodide to iodine. No characteristic peak of I₃⁻ (350 nm) was observed, which may mainly be attributed to the presence of K⁺ ions in the aqueous electrolyte, which was beneficial to the direct conversion between I⁻ and I⁰ and consistent with the DFT results in Fig. 1a (Supplementary Fig. 16 and Supplementary Note 13)[2,3,5,58]. A new strong peak (341 nm) corresponding to the formation of ICl interhalogens was observed when the system was charged to 2.4 V, verifying the conversion of I⁰ to I⁺. Furthermore, the absorption peak of the iodine molecule (375–494 nm) vanished when the system was charged to 2.4 V, consolidating the oxidation process from I⁰ to I⁺. Correspondingly, the characteristic absorption peaks almost recovered to the

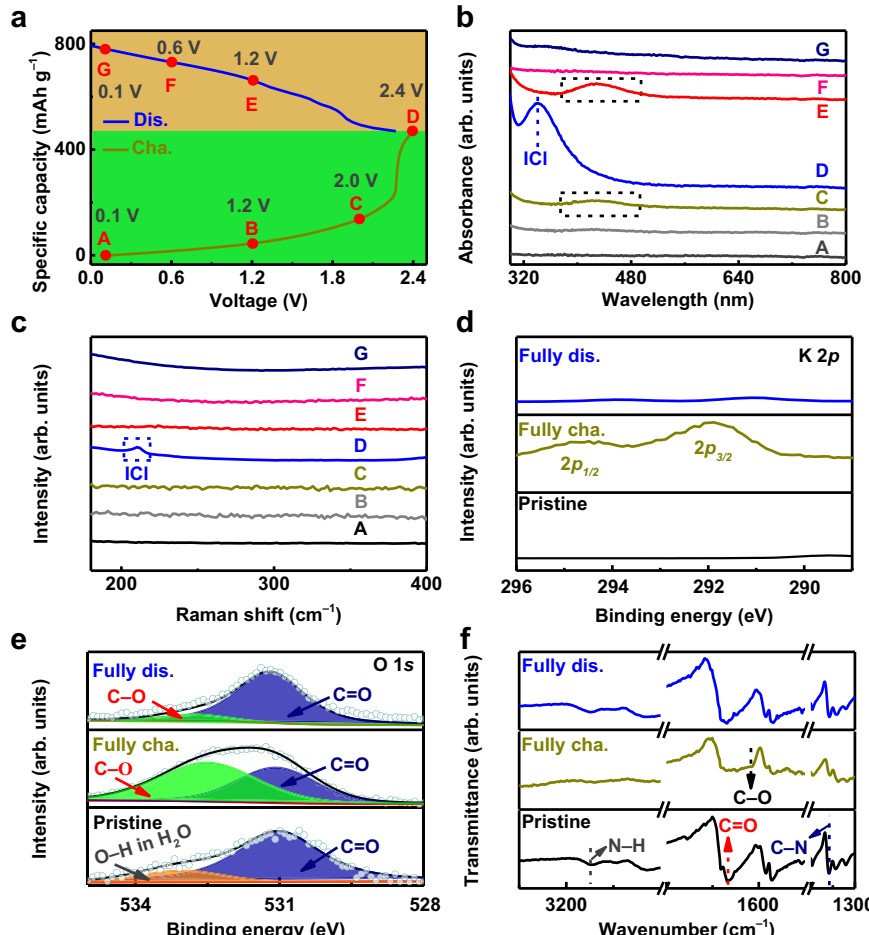

**Fig. 3 | Ex situ measurements to study the conversion of iodine in I$_2$ electrode and the K-ion storage in PTCDI electrode. a** GCD curve collected from the 10th cycle of the PTCDI‖I$_2$ glass cell tested at 40 A g$^{-1}$ and 25 ± 1 °C in a saturated KCl aqueous electrolyte with voltage points marked for characterization. **b** Ex situ UV–vis spectra measurements of the saturated KCl aqueous electrolyte recorded at various charge/discharge stages. **c** Ex situ Raman spectra measurements of the I$_2$ electrode recorded at different charge/discharge states. **d**–**f** Ex situ High-resolution K 2$p$ XPS spectra, O 1$s$ XPS spectra, and ATR–FTIR spectra of the PTCDI electrode in the pristine (uncycled), fully charged and fully discharged states, respectively.

original state during the discharge process, verifying the highly reversible conversion of I$^-$/I$^0$/I$^{+2,3,5}$. As shown in Fig. 3c, a new Raman signal (208 cm$^{-1}$) corresponding to the characteristic band of ICl interhalogens emerged when the system was charged to 2.4 V and vanished when the system was discharged to 1.2 V, consolidating the reversible I$^0$/I$^+$ conversion and in accord with the UV–vis results in Fig. 3b[2,3,5,59].

To better understand the energy storage mechanism of the PTCDI electrode, ex situ X-ray photoelectron spectroscopy (XPS) and ATR–FTIR measurements were performed at fully charged or discharge states. As illustrated in Fig. 3d, the characteristic peaks of K 2$p$ emerged in the fully charged state, while no signal peaks were detected in the pristine (uncycled) state, demonstrating the intercalation of K-ion into PTCDI. Conversely, the intensity of the K 2$p$ peaks in the fully discharged state declined sharply and almost vanished compared with that in the fully charged state, verifying the reversible deintercalation of K-ion from PTCDI[32,34,41,43]. Distinct from the pristine (uncycled) state, a new peak (532.6 eV) corresponding to the C–O species in the fully charged state was observed, verifying the conversion of C=O to C–O species (Fig. 3e)[32,42,43,60]. During the subsequent discharge process, the signal ratio of C–O/C=O decreased from 1.32 in the fully charged state to 0.12, demonstrating the reversible conversion of C–O to C=O species. In addition, the reversible intercalation/de-intercalation process of the PTCDI electrode was also confirmed by ATR–FTIR (Fig. 3f). Notably, the intensity of the stretching vibration of carbonyl groups

(–C=O) appearing at 1666 cm$^{-1}$ in the fully charged state was weaker than that in the pristine (uncycled) state. Furthermore, a new and broad peak at 1617 cm$^{-1}$ was observed, demonstrating the conversion from carbonyl groups (–C=O) to enolate groups (–C–O)[32–34,38,41–43,60]. Upon the subsequent discharge process, almost all the above characteristic peaks recovered to their original positions and intensities, suggesting the reversible conversion between carbonyl (–C=O) and enolate (–C–O) groups of the PTCDI electrode[32,33,38,41,60].

Considering the PTCDI stability toward iodine anionic species and the promising electrochemical energy storage performance of the PTCDI‖I$_2$ glass cell in a saturated KCl electrolyte, an additional amount of I$_2$(s) was introduced into the KCl electrolyte to validate the feasibility of the PTCDI‖I$_2$ glass cell in the presence of I$_2$ in the electrolyte. The operational voltage window of the saturated mixed KCl/I$_2$ aqueous electrolyte was 2.60 V, slightly broader than that of the saturated KCl electrolyte, which may be ascribed to the lower amount of free water when additional I$_2$ molecules were dissolved in the electrolyte (Supplementary Fig. 17 and Supplementary Note 14)[61].

Figure 4a displays a CV curve of the PTCDI‖I$_2$ glass cell in the saturated mixed KCl/I$_2$ aqueous electrolyte. A cascade cell reaction composed of two independent electrochemical processes (marked as step 1 and step 2) can be observed. Compared to batteries exploiting a single electrochemical reaction, battery systems with cascade reaction presents several advantages: (1) integrating two full reactions internally and avoiding the inactive additional components needed for

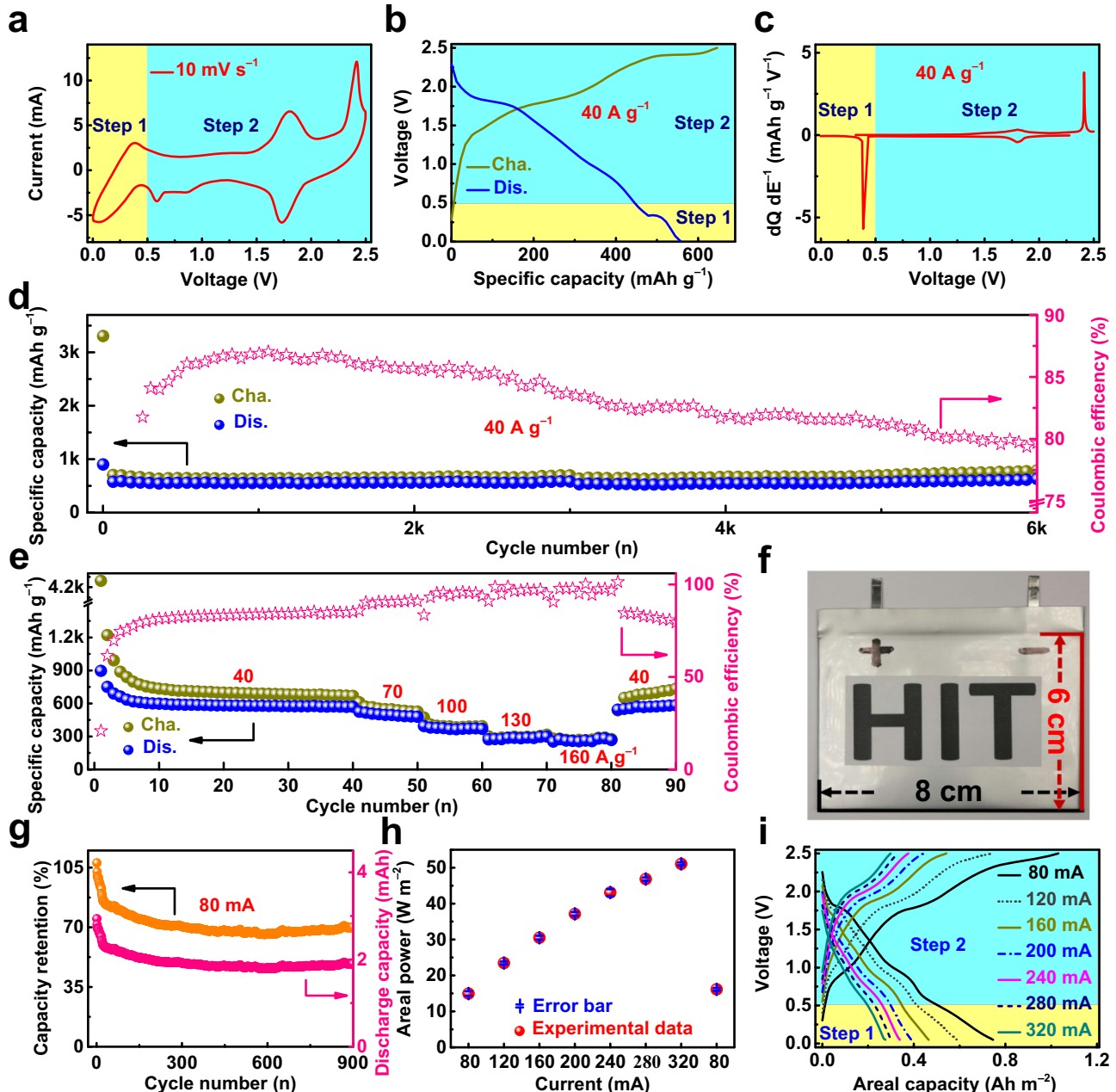

**Fig. 4 | Electrochemical performance of the PTCDI‖I₂ cascade glass cell and the PTCDI‖CP cathode-free cascade pouch cell. a** Typical CV curve of the PTCDI‖I₂ cascade glass cell in a saturated mixed KCl/I₂ aqueous electrolyte. **b**, **c** Typical GCD profile collected from the 1000th cycle and the corresponding differential capacity profile of the PTCDI‖I₂ cascade glass cell in a saturated mixed KCl/I₂ aqueous electrolyte. **d**, **e** Cycling performance and rate capability of the PTCDI‖I₂ cascade glass cell in a saturated mixed KCl/I₂ aqueous electrolyte, respectively. The specific capacity was calculated based on the mass loading of I₂ in the I₂@AC cathode. **f**–**h** Photograph, capacity retention, and rate capability of the PTCDI‖CP cathode-free cascade pouch cell in a saturated mixed KCl/I₂ aqueous electrolyte. The error bar given in **h** means the standard deviation of the three times tested results, and the values are in the range of 0.38−0.95. **i** GCD curves collected from the rate capability of the PTCDI‖CP cathode-free cascade pouch cell at various applied currents.

external connection; (2) higher utilization of the reaction chamber (a place that is mainly constructed by packaging, separator and current collector to accommodate the redox reaction of the system); and (3) higher energy output[30]. The CV curve in step 2 is consistent with that of the PTCDI‖I₂ glass cell in a pure saturated KCl electrolyte. In addition, a pair of additional redox peaks in step 1 was attributed to the reversible iodine redox processes happening in the mixed electrolyte. The carbon paper (CP, current collector in the above-discussed PTCDI‖I₂ batteries) utilized as the working electrode was investigated in a three-electrode glass cell (counter electrode: platinum foil; reference electrode: standard Ag/AgCl; electrolyte: the saturated mixed KCl/I₂

aqueous electrolyte). The tested results shown that two pairs of redox peaks corresponding to the conversion of I⁻/I⁰ and I⁰/I⁺ were observed, indicating the possibility of an I⁻/I⁰‖I⁰/I⁺ redox system self-constructed in the mixed electrolyte (Supplementary Fig. 18)[2,3,5]. Then, two pairs of redox peaks centered at different potentials can be identified (Supplementary Fig. 19a). Therefore, a pair of pronounced redox peaks within the voltage range of 0−0.5 V were detected when testing the CP‖CP glass cell, demonstrating that the I⁻/I⁰‖I⁰/I⁺ redox system self-constructs successfully in the mixed electrolyte (Supplementary Fig. 19b). Moreover, the voltage range of the I⁻/I⁰‖I⁰/I⁺ redox system was consistent with that of step 1 in Fig. 4a, further confirming that the

redox peaks in step 1 could be assigned to the reversible iodine redox processes happening in the cell. In addition, no characteristic redox peaks of the CP||CP glass cell were detected in either the saturated KCl electrolyte (Supplementary Fig. 20a) or the saturated $I_2$ (aq) electrolyte (Supplementary Fig. 20b), verifying that the $I^-/I^0||I^0/I^+$ redox system self-constructed only in the mixed electrolyte. Supplementary Fig. 20b also substantiates the importance of $Cl^-$ in the electrolyte and is in accord with the theoretical calculation results in Fig. 1b. During the charge process, the $I_2$ molecules adsorbed at the interface between the positive electrode and electrolyte were oxidized to $I^+$, while the $I_2$ molecules adsorbed at the interface between the negative electrode and electrolyte were simultaneously reduced to $I^-$ (Supplementary Fig. 21a)[2,3,5]. Conversely, all the above oxidized and reduced states of iodine recovered to their original state individually during the discharge process (Supplementary Fig. 21b).

As a part of the PTCDI||$I_2$ cascade cell system, the electrochemical performance of the $I^-/I^0||I^0/I^+$ redox system (step 1) was systematically evaluated by testing the CP||CP glass cell in the mixed electrolyte (3.4 mol $l^{-1}$ KCl + 0.0016 mol $l^{-1}$ $I_2$ in water solvent) (Supplementary Fig. 22). It is noted that the discharge areal capacity and areal current density values of the CP||CP glass cell were calculated based on the total area (-2 $cm^2$) of both sides of the positive electrode. A charge plateau at 0.38 V and a discharge plateau at 0.19 V were observed in the GCD curve of the $I^-/I^0||I^0/I^+$ redox system (Supplementary Fig. 22a), in accord with the CV results in Supplementary Fig. 19b. A discharge areal capacity of 58.7 µAh $cm^{-2}$ was obtained at 0.2 mA $cm^{-2}$ after 9800 cycles. The CP||CP glass cell discharge capacity increased during cycling due to the increasing number of activated $I_2$ molecules. The relatively low and fluctuant CE values were mainly attributed to the electrochemical reaction of the symmetric cell occurring at the interface between the electrode and electrolyte[9,62–65]. On the one hand, the reactant and product of the symmetric cell were soluble in the electrolyte, leading to the relatively low CE values. On the other hand, the iodine molecules were electroneutral, and only the iodine molecules physically adsorbed on the electrode participated in the redox reaction, thus causing fluctuation of the CE values (Supplementary Fig. 22b). Moreover, discharge areal capacities of 30, 16, 8 and 6 µAh $cm^{-2}$ were achieved at 2, 5, 10 and 15 mA $cm^{-2}$, respectively, indicating a good rate capability (Supplementary Fig. 22c). A new UV−vis peak centered at 343 nm corresponding to ICl interhalogens emerged in the fully charged state and almost vanished in the fully discharged state, verifying the reversible conversion of $I^0/I^+$ (Supplementary Fig. 22d)[2,3,5]. In view of the large amount of $I_2$ contained in the mixed electrolyte, the intensity change of the broad absorption peak (375−494 nm) corresponding to $I_2$ molecules was not apparent in different states of charge[2,3,5].

As presented in Fig. 4b, a major discharge plateau at 0.34 V corresponding to the $I^-/I^0||I^0/I^+$ redox system was clearly observed in step 1, and the GCD curve in step 2 was identical to that of the PTCDI||$I_2$ glass cell in the saturated KCl electrolyte, as shown in in Fig. 2b. Moreover, these results are in accord with the CV results in Fig. 4a and confirm the successful construction of the PTCDI||$I_2$ cascade glass cell in the mixed electrolyte. In addition, the differential capacity analysis was carried out to evaluate the reactions of the PTCDI||$I_2$ cascade glass cell in the mixed electrolyte. As shown in Fig. 4c, two peaks at 1.81 V and 0.38 V during the discharge process are observed in accord with the results in Fig. 4a, b, consolidating the successful construction of the PTCDI||$I_2$ cascade glass cell in the mixed electrolyte. Overall, the first charge step (marked as step 1) of the cascade cell based on PTCDI||$I_2$ in the saturated mixed KCl/$I_2$ aqueous electrolyte corresponds to the conversion of the $I^-/I^0||I^0/I^+$ redox system within the voltage range of 0−0.5 V. That is, the conversion of $I^-/I^0$ occurred at the negative electrode and the conversion of $I^0/I^+$ occurred at the positive electrode simultaneously. During the second charge step (marked as step 2) within the voltage range of 0.5−2.5 V, the conversion of $I^-/I^0||I^0/I^+$ occurred at the positive

electrode, and the intercalation of K-ion occurred at the PTCDI electrode (Supplementary Fig. 23a−c)[2,3,5]. The kinetic behaviors of the cascade cell were evaluated under various scan rates, and the whole redox process was found to be regulated by the combination of diffusion-controlled and capacitance-dominated effects (Supplementary Fig. 24a, b and Supplementary Note 15). The cascade cell displays a large capacitive contribution under high scan rates (Supplementary Fig. 24c). In addition, in situ pressure test during the CV measurement of the PTCDI||$I_2$ cascade glass cell in the saturated mixed KCl/$I_2$ aqueous electrolyte was performed to evaluate the stability of the mixed electrolyte. The tested results show that no fluctuation in gas pressure was detected during the whole process of the cascade cell, which indicate that no $H_2$ or $O_2$ evolution reaction occurred and are in accord with the results in Supplementary Fig. 17, consolidating the electrochemical stability of the mixed electrolyte within 2.5 V (Supplementary Fig. 25)[61]. As shown in Fig. 4d, a discharge capacity of 628 mAh $g^{-1}$ of the PTCDI||$I_2$ cascade glass cell was delivered under the specific current of 40 A $g^{-1}$ at 25 ± 1 °C after 6000 cycles, and an average cell discharge voltage of 1.25 V was obtained. Besides, the CE values fluctuated owing to the inclusion of the $I^-/I^0||I^0/I^+$ redox system. The PTCDI||$I_2$ cascade glass cell was able to sustain 105,000 cycles at 60 A $g^{-1}$ and 25 ± 1 °C, delivering a final specific discharge capacity of about 69 mAh $g^{-1}$ (Supplementary Fig. 26). The morphology and particle size of the PTCDI electrode after cycling at 40 A $g^{-1}$ for 9200 cycles in the mixed electrolyte were very similar to those of the uncycled electrodes, thus confirming the structural and (electro)chemical stability of the PTCDI-based electrodes upon prolonged cycling at high currents (Supplementary Fig. 27). As illustrated in Fig. 4e, discharge capacities of 574, 500, 376, 285 and 264 mAh $g^{-1}$ of the PTCDI||$I_2$ cascade glass cell at 25 ± 1 °C were delivered at 40, 70, 100, 130 and 160 A $g^{-1}$, respectively. Notably, a discharge capacity of 572 mAh $g^{-1}$ was delivered when the specific current was shifted from 160 to 40 A $g^{-1}$, the value of which is 99.7% of that at the initial 40 A $g^{-1}$, demonstrating an efficient rate capability of the PTCDI||$I_2$ cascade glass cell. In addition, compared with the previously reported aqueous batteries (0−3.2 V), the operational voltage range (0−2.5 V) of this cascade glass cell was among top values (Supplementary Fig. 28 and Supplementary Note 16). Therefore, the PTCDI||$I_2$ cascade glass cell in mixed electrolyte demonstrates the feasibility of the cascade cell in $I_2$-based aqueous batteries, making full use of the iodine coming from the $I_2$ cathode and outputting more energy as an $I^-/I^0||I^0/I^+$ redox system in the electrolyte.

Inspired by the results that the conversion of $I^-/I^0||I^0/I^+$ can be completed by using a CP as the electrode in the mixed electrolyte (Supplementary Fig. 18), an aqueous cathode-free cascade glass cell of PTCDI||CP was successfully constructed in the same mixed electrolyte (Supplementary Fig. 29 and Supplementary Note 17). Owing to the same reaction mechanism, the reaction processes of the PTCDI||CP cathode-free cascade glass cell are identical to those of the PTCDI||$I_2$ cascade glass cell (Supplementary Fig. 29a−f). Similarly, the PTCDI||CP cathode-free cascade glass cell demonstrated stable cycling performance and good rate capability (Supplementary Fig. 29g−i). Furthermore, a large-area pouch cell of the PTCDI||CP cathode-free cascade cell (the pouch cell was made of 1 single-layer coated PTCDI electrode) with dimensions of 6 cm × 8 cm was successfully assembled (Fig. 4f). The capacity retention reached 70% at 80 mA after cycling for 900 cycles, delivering a final discharge capacity of about 1.91 mAh and exhibiting good cyclical stability (Fig. 4g). An areal power of 52 W $m^{-2}$ was achieved when a current of 320 mA was applied, indicating the rate tolerance of the cascade cell. In addition, an areal power of 16 W $m^{-2}$ was achieved when the current was shifted from 320 to 80 mA, the value of which was almost identical to that at the initial 80 mA, exhibiting good reversibility (Fig. 4h). As shown in Fig. 4i, the GCD curves of the PTCDI||CP cathode-free cascade pouch cell at 80 mA exhibit three discernible plateaus during the discharge process and correspond to reaction mechanism of the cascade cell, and the discharge

plateaus turned sloped under high currents owing to the increased polarization[66].

## Discussion

In summary, we have designed, assembled and tested aqueous PTCDI‖$I_2$ batteries with aqueous saturated KCl-based electrolyte solutions. The electrodes demonstrated structural and (electro)chemical stability during cell cycling also delivering in full cell configuration a lifespan of up to 92,000 cycles at a specific current of 40 A g$^{-1}$ and 25 ± 1 °C. In addition, the maximum calculated specific energy and power values were 434 Wh kg$^{-1}$ at 50,420 W kg$^{-1}$ and 155,072 W kg$^{-1}$ at 86 Wh kg$^{-1}$ (these values were obtained in the initial cycles), respectively. Furthermore, considering the characteristics of intrinsic inertness to various iodine anionic species of PTCDI, the introduction of $I_2$ into the electrolyte was favorable for the PTCDI‖$I_2$ battery, leading to successful self-construction of a cascade cell in a mixed KCl/$I_2$ aqueous electrolyte. The cascade cell can further reach 2.5 V (the high cut-off voltage) and work for 105,000 cycles at 60 A g$^{-1}$. Moreover, in view of the results that the conversion of I$^-$/I$^0$/I$^+$ can be completed when using only a CP current collector as the electrode in the mixed electrolyte, the PTCDI‖CP cathode-free cascade cell was assembled and tested. In addition, a 6 cm × 8 cm PTCDI‖CP pouch cell was also assembled and tested, demonstrating good long-term electrochemical energy storage capabilities.

## Methods

### Materials

Iodine ($I_2$; ≥ 99.8%, Aladdin, China), potassium chloride (KCl; 99%, Aladdin, China), the standard Ag/AgCl electrode (CHI111, Chenhua, China), carbon paper (CP; 0.2 mm thickness; 1.5 ± 0.05 g cm$^{-3}$; tensile strength ≥3.5 MPa; Jinglong Special Carbon Technology Co., LTD; China), 3,4,9,10-perylenetetracarboxylic diimide (PTCDI; 95%, Alfa Aesar, America), and active carbon (AC; XFNANO, China) were purchased and used without further treatment. Zinc (Zn) foil (0.1 mm thickness, ≥99.99%) was purchased from Tianjin Annohe New Energy Technology Co., LTD (China) and used without further treatment. The platinum (Pt) foil electrode (≥99.99%, 1 cm × 1 cm) and the Pt sheet electrode clip (99.99%) were purchased from Shanghai Jingchong Electronic Technology Development Co., LTD (China) and used without further treatment.

### Synthesis

$I_2$@AC was synthesized through a facile method. Briefly, 1 g $I_2$ and 1 g AC were mixed by grinding using an agate mortar in the air at 25 ± 1 °C for 3 min. Then, the mixed powder was sealed in a 10 ml hydrothermal polytetrafluoroethylene reactor (Suzhou Shenghua Instrument Technology Co., LTD; China) and heated at 80 °C for 4 h. After natural cooling, active carbon-enveloped $I_2$ ($I_2$@AC) was obtained.

### Preparation of electrodes

To prepare the PTCDI-based negative electrode, PTCDI, Ketjen Black (KB; 99%; 48 μm; EC300J, Lion, Japan) and polyvinylidene fluoride (PVDF; ≥99.5%; PVDF5130, Guangdong Canrd New Energy Technology Co., LTD.; China) binder were homogeneously mixed in solids under stirring for 15 min at a weight ratio of 7:2:1. Then, N-methyl pyrrolidone (NMP) solvent was added into the above mixture and the newly obtained mixture was stirred for 4 h to form a homogeneous slurry. Subsequently, the slurry was coated onto a carbon paper (CP; 0.2 mm thickness; 1.5 ± 0.05 g cm$^{-3}$; tensile strength ≥3.5 MPa) substrate and the coated area was about 4–4.5 cm$^2$. Finally, the electrode was obtained after being dried under vacuum at 80 °C for 12 h and no calendaring process was applied. The average thickness and the average mass loading of the PTCDI electrode were 0.15 mm and 1.0–1.2 mg cm$^{-2}$, respectively.

To prepare the $I_2$-based positive electrode, $I_2$@AC, KB and PVDF binder were homogeneously mixed in solids under stirring for 15 min

at a weight ratio of 8:1:1. Then, N-methyl pyrrolidone (NMP) solvent was added into the above mixture and the newly obtained mixture was stirred for 4 h to form a homogeneous slurry. Subsequently, the slurry was coated onto a CP (0.2 mm thickness; 1.5 ± 0.05 g cm$^{-3}$; tensile strength ≥3.5 MPa) substrate and the coated area was about 0.8–1.0 cm$^2$. Finally, the electrode was obtained after being dried under vacuum at 40 °C for 12 h and no calendaring process was applied. The average thickness and the average mass loading of the $I_2$ electrode were 0.15 mm and 1.0–1.2 mg cm$^{-2}$, respectively.

To prepare the AC electrode, AC, KB and PVDF binder were homogeneously mixed in solids under stirring for 15 min at a weight ratio of 8:1:1. Then, N-methyl pyrrolidone (NMP) solvent was added into the above mixture and the newly obtained mixture was stirred for 4 h to form a homogeneous slurry. Subsequently, the slurry was coated onto a CP (0.2 mm thickness; 1.5 ± 0.05 g cm$^{-3}$; tensile strength ≥3.5 MPa) substrate and the coated area was about 0.8–1.0 cm$^2$. Finally, the electrode was obtained after being dried under vacuum at 80 °C for 12 h and no calendaring process was applied. The average thickness and the average mass loading of the AC electrode were 0.15 mm and 1.0–1.2 mg cm$^{-2}$, respectively.

To prepare the KB electrode, KB and PVDF binder were homogeneously mixed in solids under stirring for 15 min at a weight ratio of 9:1. Then, N-methyl pyrrolidone (NMP) solvent was added into the above mixture and the newly obtained mixture was stirred for 4 h to form a homogeneous slurry. Subsequently, the slurry was coated onto a CP (0.2 mm thickness; 1.5 ± 0.05 g cm$^{-3}$; tensile strength ≥3.5 MPa) substrate and the coated area was about 0.8–1.0 cm$^2$. Finally, the electrode was obtained after being dried under vacuum at 80 °C for 12 h and no calendaring process was applied. The average thickness and the average mass loading of the KB electrode were 0.15 mm and 1.0–1.2 mg cm$^{-2}$, respectively.

### Cell fabrication

A PTCDI‖$I_2$ glass cell was assembled with a PTCDI electrode (2 × 3 cm) as the negative electrode and an $I_2$@AC electrode (1 × 3 cm) as the positive electrode in a mass ratio of 4:1 (the negative electrode/the positive electrode), and 10 ml saturated KCl aqueous solution was used as the electrolyte. The distance between the two electrodes was ~1 cm. Similarly, a PTCDI‖$I_2$ cascade glass cell was assembled with 10 ml saturated mixed KCl/$I_2$ aqueous solution as the electrolyte. A cathode-free PTCDI‖CP cascade glass cell was assembled with CP (1 × 3 cm) as the positive electrode and 10 ml saturated mixed KCl/$I_2$ aqueous solution as the electrolyte. The Zn‖$I_2$ glass cell was assembled with a Zn foil (2 × 3 cm) as the negative electrode and an $I_2$@AC electrode (1 × 3 cm) as the positive electrode, and a 10 mL1 M KCl + 1 M ZnCl$_2$ mixed solution as electrolyte. The self-discharge rate of the Zn‖$I_2$ glass cell was tested at the open circuit after charging to 1.8 V at 40 A g$^{-1}$. A photographic picture of the typical assembled two-electrode glass cell configuration is presented in Supplementary Fig. 30. The three-electrode glass cell was assembled with the working electrode (KB electrode, PTCDI electrode, CP electrode or $I_2$@AC electrode), the Pt foil (≥99.99%, 1 cm × 1 cm) counter electrode and standard Ag/AgCl reference electrode (CHI111, Chenhua, China) in a 10 ml aqueous electrolyte (Supplementary Fig. 31). All of the glass cells mentioned above were hermetically sealed by laboratory paper film (PM996, Bemis, America) to guarantee that electrodes were immersed into the electrolyte during the whole test and no crystallization of electrolyte was observed. The pouch-type cathode-free PTCDI‖CP cascade cell was assembled with a stacked three-layer. Specifically, the single-side coated PTCDI electrode (8 × 6 cm), glass microfibre filter separator (Whatman, GF/D; 8 × 6 cm) fully infiltrated in a saturated mixed KCl/$I_2$ aqueous electrolyte (4 ml) and CP electrode (8 × 6 cm) were compacted tightly and sealed by aluminum plastic film at a dry room (the dew point was 15 °C) under ambient conditions. The pressure and temperature applied to the cell during cycling were 101.325 kPa and 25 ± 1 °C, respectively.

## Physicochemical characterizations

Ex situ X-ray diffraction (XRD; Ultima IV, Rigaku, Japan) measurements using Cu Kα radiation were carried out to determine the structure and phase composition and the diffraction data was collected at a step mode (1° min$^{-1}$) over the angular range of 5–60°. Ex situ Attenuated Total Reflection Flourier transformed infrared spectroscopy (ATR–FTIR; Nicolet iS5, Thermo Scientific, America) measurements were performed to determine the functional groups of organic molecules. Brunauer–Emmett–Teller (BET; ASAP 2460, Quantachrome, America) surface areas were measured using N$_2$ adsorption–desorption at 77 K. Samples were dried for 24 h in the vacuum oven at 40 °C prior to the characterization. Thermogravimetric analysis (TGA) measurements were performed to determine the content of iodine with a TG-DSC analyser (NETZSCH, STA 449 F3, Germany), and the heating rate was 5 °C min$^{-1}$ from 25 ± 1 °C to 500 °C under an Ar atmosphere. Ex situ X-ray photo-electron spectroscopy (XPS; PHI 5000 Versaprobe III, Ulvac–Phi, Japan) with a monochromic Al Kα X-ray source and an Ar ion cluster sputtering gun was performed to analyze the surface composition and valance evolution details. All spectra were calibrated using the binding energy of C 1$s$ (284.8 eV) as a reference. Field emission scanning electron micro-scopy (FE–SEM; S4800, HITACHI, Japan; 10 kV, 5 mA) was employed to characterize the morphology. Ex situ Ultraviolet–visible (UV–vis; Lambda 365, Perkin Elmer, America) spectroscopy measurements were carried out over a range from 200 to 800 nm. Ex situ Raman spectro-scopy measurements were carried out on a bench Raman dispersive microspectrometer (Qontor, Renishaw/InVia, England) with a 532 nm laser at frequencies from 100 to 4000 cm$^{-1}$ to record the crystal-lographic information. The electrode samples for ex situ measurements were collected by disassembling the glass cell and were washed by deionized water to remove the residual electrolyte, and were dried in a vacuum oven at 40 °C for 12 h finally. A sample holder with an inert atmosphere was used to transport the electrode samples from the Ar-filled glovebox to the equipment used for the ex situ measurements.

## Electrochemical characterization

Cyclic voltammetry (CV) tests and linear sweep voltammetry (LSV) tests, and electrochemical impedance spectroscopy (EIS) measurements were carried out on an electrochemical workstation (CHI760E, Chenhua, China). The EIS measurements were carried out with a potential amplitude of 5 mV and 12 points per decade of frequency in the range from 10$^{-2}$ Hz to 10$^{5}$ Hz. The open-circuit voltage time applied before carrying out the EIS measurement was 5 min. The electrochemical potential window of the aqueous electrolyte was determined through a three-electrode glass cell system (working electrode: KB electrode; counter electrode: platinum foil electrode (≥99.99%, 1 cm × 1 cm); reference electrode: standard Ag/AgCl electrode (CHI111, Chenhua, China)) (Supplementary Fig. 31). The CV tests or GCD tests of the PTCDI electrode, KB electrode, CP electrode and I$_2$@AC electrode were carried out in a three-electrode glass cell at 25 ± 1 °C. Galvanostatic charge/discharge (GCD) tests and cycling performance tests (rest for 10 h after the assembling of cell, and then charge to the high cut-off voltage and discharge to the low cut-off voltage under galvanostatic condition without intermediate rest), and rate capability tests were conducted in a battery test system (CT3002A, LANHE, China). An in situ pressure test during the CV test was performed to determine the H$_2$ and O$_2$ evolution through the pressure transducer (0–2 bar (A), 12–36 VDC; CYYZ11, Starsensor, China) (Supplementary Fig. 32). The electrochemical energy storage tests were carried out in an environmental chamber (25 ± 1 °C). The specific current was calculated based on the mass of I$_2$ in I$_2$@AC. The specific capacity $C$ (mAh g$^{-1}$) was calculated as follows[67]:

$$C = \frac{I\Delta t}{3.6m} \quad (5)$$

where $I$ (A) is the applied current, $\Delta t$ (s) is the corresponding charge or discharge time, and $m$ (g) is the weight of I$_2$ in the positive electrode.

The specific energy ($E$, Wh kg$^{-1}$) and corresponding specific power ($P$, W kg$^{-1}$) were calculated as follows[33]:

$$E = \frac{\int IV(t)dt}{3.6m} \quad (6)$$

$$P = \frac{3600E}{t} \quad (7)$$

Where, $I$ (A) is the applied current, $V$ (V) is the voltage of the cell, $t$ (s) is the corresponding discharge time, and $m$ (g) is weight of I$_2$ in the positive electrode.

The average discharge voltage (V) was calculated as follows[68]:

$$\text{Average discharge voltage (V)} = \frac{\text{Specific energy}\left(\text{Wh kg}^{-1}\right)}{\text{Specific capacity}\left(\text{Ah kg}^{-1}\right)} \quad (8)$$

## Computational details

All structures were optimized by density functional theory (DFT) using the B3LYP functional with the LANL2DZ basis set, and using the Gaussian 09 W program[69]. All calculations were carried out with atom-pairwise dispersion correction (DFT-D3) and the implicit universal solvation model based on the solute electron density (SMD)[70].

To clarify the mechanism of selected cations on the conversion of I$^-$/I$^0$, several systems were simulated, and the Gibbs free energy change ($\triangle G$) was used to measure the degree of spontaneity of different sys-tems (formula shown below):

$$\triangle G = G_{\text{product}} - G_{\text{reactant}}$$

The dissociation energy ($E_{\text{dissociation}}$) was computed from:

$$E_{\text{dissociation}} = E_{\text{anion}} + E_I - E_{\text{group}}$$

$E_{\text{group}}$, $E_I$ and $E_{\text{anion}}$ are the energies of the group, I$^+$ ion and iso-lated anions, respectively.

## Reporting summary

Further information on research design is available in the Nature Research Reporting Summary linked to this article.

# Data availability

The detailed data generated in this study are provided in the Source Data file. Source data are provided with this paper.

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

## Acknowledgements
This research was supported by the Project of International Science and Technology Cooperation in Guangdong Province (No. 2020A0505100016, Y.H.), the Shenzhen Sauvage Nobel Laureate Laboratory for Smart Materials (Y.H.), the Shenzhen Science and Technology Innovation project (fundamental 20220308, Y.H.), the Shenzhen Science and Technology Program (No. KQTD20200820113045083, Y.H.; No. ZDSYS20190902093220279, Y.H.) and State Key Lab of Advanced Welding and Joining, Harbin Institute of Technology (Y.H.).

## Author contributions
Y.H., M.Y. and Z.Z. designed the research. Z.Z. conducted material preparations, electrochemical tests and characterizations. Y.Z., Y.J. and Z.Z. conducted the computations. Z.Z., Y.H., M.Y. and Y.J. prepared the manuscript. All authors contributed to the discussion of the data.

## Competing interests
The authors declare no competing interests.
