## [Peer Review File · Nature Communications]

REVIEWER COMMENTS

Reviewer #1 (Remarks to the Author):

Review report on: Aqueous organic anode//iodine cascable battery with ultra-long lifespan, high energy and power density

The authors report on a novel aqueous PTCDI//I₂ cascade battery in a saturated KCl and I₂(aq) mixed electrolyte. The presented results show very high energy and power densities in combination with a very long cycle life of this system.

In Summary, this work is very interesting to specialists within the field of battery research. The presented experimental data and methodology is sound and said performance is, to the best of my knowledge, outstanding in the field. I therefore can recommend its publication in the Journal of Nature Communications upon minor corrections.

Minor Comments:

The level of English needs some improvement concerning typos, gramma and word order before it can be accepted for publication. Especially definite and indefinite articles are missing on several occasions in the manuscript.

Page 17, line 302: "...capacities of 0.03, 0.016, 0.008 and 0.006 mAh cm⁻² were achieved" The authors should present the values in microampere hours for better readability.

Page 17, line 303: "A new peak centered at 343 nm corresponding to ICl interhalogens emerged..." The authors should add, that this peak relates to UV-vis measurements, additionally the peak in Fig S13f is barely visible. The authors should increase the y-axis, perhaps by plotting all three spectra in one graph.

The authors show impressive performance at high charge/discharge currents, i.e. 40 A g⁻¹, how does the cell behave at low currents i.e. 1 A g⁻¹? Are there any results on the self-discharge rate of the full cell?

Reviewer #2 (Remarks to the Author):

This paper describes a very interesting concept of new type of electrochemical cell based on cheap materials. The authors were able to come up with explanations and simple models to analyse and understand the measured data. In that respect they used a number of relevant analytical tools to proof the behaviour and existence of the electro-active species. Unfortunately, the paper is quite chaotic, and actually it is way too difficult to read – that is why it took me a little longer to go through. Figures are complicated with often irrelevant information, whereas important figures are placed in the supporting information, where they do not belong. The paper also lacks of a simple overall schematic of the cascade cell from where they authors can start to explain their work and results – this would absolutely help. In a similar aspect, why mentioning the important electrochemical steps for the first time in the captions of Figure 4, and then only at line 314 – this could for instance govern the schematic figure in the beginning. On the other hand, Fig. 3 is a figure where we can indeed read relevant data, important for the understanding of the story, this is how it should look like.

With respect to the data evaluation, I have to following comments:

Regarding the current evaluation in the CV curves, wouldn't it be better to have deconvolution of the various peaks as the peaks are quite broad and processes are overlapping. In the same respect, in Fig s3b, peak 4 may consist of two peaks, similarly in Fig s7b, where peak 3 obviously has composed of at least two peaks. Besides, the width of the scan rates is small, i.e. only a factor of 2 and 4 respectively – it is questionable whether this is enough to use the underlying model to conclude on the values given.

Other points:

- In line 122 of the supporting information, you gave a reaction with K⁺, but the product formed has also a proton accepted (middle product) – was wondering whether this is a typo or are protons indeed involved in this reaction?

- In Fig 2f, a cycle number of 90k is given meaning 90,000. This seems to be a lot: hence considering a 10C charge and discharge, a full cycle takes 12 minutes 0.2 hours. Every year has 8760 hours, which means at a charge/discharge of 10C it 90,000 cycles will take just more than 2 years. Hence, was wondering whether this is indeed the case – or at least provide the C-rate somewhere.

In summary: Please reorganise the paper so that the readers are being triggered, and not to have figuring out themselves where the relevant information has been placed. This too, includes selecting the important figures – which one to put in the body of the paper, and which in the supporting information. With respect to data analysis, reanalysing and deconvolution of the CV curves would be essential to better calculate the fractions of the capacitive-controlled vs the diffusive-controlled behaviour.

Reviewer #3 (Remarks to the Author):

The cell voltage of the proposed organic cathode//I₂ cascade battery is 2.5 V, which is well above the thermodynamic potential of the water splitting. The authors provided the LSV data to support the there is no water splitting. But it is not convincing.

There is no mention of percentage of specific capacity retention against the theoretical specific capacity of the cell. Figure 2F, no explanation for linear decrease in specific capacity with cycle number.

The oxidative stability of PTCDI is questionable (<https://doi.org/10.1002/chem.202003624>).

There was no formation of I₃⁻, as supported from the UV-visible spectra, XPS, Raman and FTIR. What could be the possible reason?

Line no. 185-189, the explanation is repeated.

Can author provides the details about specific capacity calculations?

The quantitative adsorption of I₂ on activated carbon is not mentioned.

Can author provide the kinetic details of the cascade reactions?

Figure 2C and 2D, CV and alpha graph, it is not clear about the practical capacity of the battery. The high capacity was due to the dip discharging. Please explain.

Figure S3 D, shows the capacitive contribution of the electrode is very high compare to the diffusion controlled. Could you please explain it's contribution to specific capacity of the battery.

Figure S7, shows the shift in peak potential of the full cell voltammogram of PTCDI//I₂ in KCl solution. Also it is showing two cathodic peaks after 40th and 90th scan. The explanation provided in line no. 149 is insufficient.

Figure S10, CV shows substantial increase in ohmic resistance of the PTCDI electrode after 3rd, 4th and 5th scan. Explanation is not provided.

Dear Reviewers,

We would like to thank you for taking time to evaluate our submission (Manuscript ID: NCOMMS-22-09631) and raising the positive remarks and constructive comments, which have helped us substantially improved the quality of our manuscript. We have thoroughly addressed all comments/concerns from the reviewers point-by-point with the associated changes listed in this letter and highlighted in yellow in the revised manuscript and supplementary information. We sincerely hope that we have successfully dealt with all the questions so that you would find our revised manuscript publishable in Nature Communications.

Yours faithfully,

Yan Huang

Reviewer #1 (Remarks to the Author):

Comments:

The authors report on a novel aqueous PTCDI||I₂ cascade battery in a saturated KCl and I₂(aq) mixed electrolyte. The presented results show very high energy and power densities in combination with a very long cycle life of this system. In Summary, this work is very interesting to specialists within the field of battery research. The presented experimental data and methodology is sound and said performance is, to the best of my knowledge, outstanding in the field. I therefore can recommend its publication in the Journal of Nature Communications upon minor corrections.

Answer: We highly appreciate the reviewer for raising the constructive comments and positive remarks on our manuscript. Following this reviewer's suggestions, we have carried out additional experiments and made revisions in the revised manuscript and Supplementary Information (SI). Please refer to our point-by-point response below.

Minor Comments:

--- The level of English needs some improvement concerning typos, gramma and word order before it can be accepted for publication. Especially definite and indefinite articles are missing on several occasions in the manuscript.

Answer: We are grateful to the reviewer for pointing out this issue. Following this reviewer's suggestion, the manuscript and SI have been re-edited by Nature Research Editing Service (Certificate ID: WLJ64DBT). Please see the details in the revised manuscript (Page 2, Line 23; Page 3, Line 52-60, 61-62, 65-66, 68-69, 72, 74; Page 4, Line 79-80, 83, 89-93, 95, 99-102; Page 5, Line 107, 110, 112-117, 119, 130-133; Page 6, Line 140; Page 7, Line 160; Page 8, Line 166, 168, 170-173, 179; Page 9, Line 195-196, 205-206, 213-214; Page 10, Line 230, 234, 240-242, 245, 247; Page 12, Line 251, 254, 256, 264, 270-272, 276-277; Page 13, Line 281-286, 288, 291-292; Page 14, Line 299-300, 302, 306; Page 15, Line 308, 311-312, 318, 326, 331, 334, 336; Page 16, Line 345-346, 363, 365; Page 17, Line 368-369, 383, 387-388, 392, 395-396; Page 18, Line 398, 402, 404, 408, 414; Page 19, Line 416, 419; Page 20,

Line 423, 424, 428, 435, 437, 441-442, 449; Page 23, Line 519-520, 522, 525, 527) and SI (Page 3, Line 25, 30-31; Page 5, Line 53-54, 56, 58, 60, 64; Page 7, Line 109; Page 10, Line 142, 147; Page 11, Line 152; Page 16, Line 217; Page 27, Line 311; Page 29, Line 326, 330, 332; Page 30, Line 343; Page 31, Line 350; Page 32, Line 353).

--- Page 17, line 302: "...capacities of 0.03, 0.016, 0.008 and 0.006 mAh cm⁻² were achieved" The authors should present the values in microampere hours for better readability.

Answer: We thank the reviewer for raising the constructive comment. Following the reviewer's suggestion, we have modified the sentence in the revised manuscript (Page 16, Line 359-361) as follows:

"Moreover, discharge areal capacities of 30, 16, 8 and 6 μAh cm⁻² were achieved at 2, 5, 10 and 15 mA cm⁻², respectively, indicating a good rate capability (Supplementary Figure 17c)."

--- Page 17, line 303: "A new peak centered at 343 nm corresponding to ICl interhalogens emerged..." The authors should add, that this peak relates to UV-vis measurements, additionally the peak in Fig S13f is barely visible. The authors should increase the y-axis, perhaps by plotting all three spectra in one graph.

Answer: We thank the reviewer for raising the constructive comment. Following this reviewer's suggestion, we have rewritten the sentence in the revised manuscript (Page 16, Line 361-363) and modified Supplementary Figure 13f (the new Supplementary Figure 17d) as follows:

"A new UV-vis peak centred at 343 nm corresponding to ICl interhalogens emerged in the fully charged state and almost vanished in the fully discharged state, verifying the reversible conversion of I⁰/I⁺ (Supplementary Figure 17d).^{2, 3, 5}"

Supplementary Figure 17. d, UV-vis spectra of the utilized electrolyte recorded in the pristine, fully charged and fully discharged states.

--- The authors show impressive performance at high charge/discharge currents, i.e. 40 A g^{-1} , how does the cell behave at low currents i.e. 1 A g^{-1} ? Are there any results on the self-discharge rate of the full cell?

Answer: We are grateful to the reviewer for raising the important points. Following this reviewer's suggestion, we have evaluated the performance of the full cell at low current density (1 A g^{-1}). The results revealed that the cell was unable to be charged to 2.4 V but only fluctuated in the vicinity of 1.8 V. This is consistent with the previous results (*Chem. Eur. J.* 2020, **26**, 17559-17566), showing that PTCDI electrode is not suitable for working at the low current density. Such phenomenon is common for organic electrodes, which is mainly attributed to the irreversible side reactions induced at a low current density and the limited intrinsic conductivity of organic electrode materials (*Chem. Eur. J.* 2020, **26**, 17559-17566; *Nat. Energy* 2019, **4**, 495-503; *Nat. Commun.* 2021, **12**, 2400; *Nat. Sustain.* 2022, **5**, 225-234; *ACS Nano* 2021, **15**, 1077-1085). In this regard, a high current density is necessary to reach the required voltage and to prevent the irreversible side reactions. The results have been added in Supplementary Figure 10 and addressed accordingly in the revised manuscript (Page 9, Line 193-195) as follows:

“Due to the inevitable side reactions at a low current density (Supplementary Figure 10), a high current density was required to reach a certain voltage and avoid irreversible side reactions.”^{32, 33, 48, 49}

Supplementary Figure 10. GCD curve of the PTCDI|| I_2 full cell in a saturated KCl electrolyte at 1 A g^{-1} .

Following this reviewer's suggestion, we have also measured the self-discharge rate of the PTCDI|| I_2 full cell after charging it to 2.4 V at 40 A g^{-1} . In fact, self-discharge behavior is common for the aqueous iodine cathode battery systems. The self-discharge is originated from the dissolubility of iodine and iodine species in the aqueous environment of the batteries (*Energy Environ. Sci.* 2021, **14**, 407-413; *Angew. Chem. Int. Ed.* 2021, **60**, 12636-12647; *Nat. Commun.* 2021, **12**, 170; *Angew. Chem. Int. Ed.* 2022, **61**, e202113576; *Adv. Mater.* 2021, **33**,

2006897; *Nano Lett.* 2015, **15**, 5982-5987; *ACS Energy Lett.* 2017, **2**, 2674-2680). This means that regardless of which kind of iodine cathode material is utilized, iodine species will inevitably release into the aqueous electrolyte and trigger a series of complexation reactions, thus leading to the self-discharge behavior. To better evaluate the self-discharge rate of the full cell (PTCDI||I₂) of this work, we constructed a Zn||I₂@AC full cell (a conventional aqueous iodine full cell that has been reported extensively, e.g. *Energy Environ. Sci.* 2021, **14**, 407-413; *Angew. Chem. Int. Ed.* 2021, **60**, 12636-12647) in mixed electrolyte (1 M KCl and 1M ZnCl₂) and evaluated its self-discharge performance after charging it to 1.8 V at 40 A g⁻¹, as a control group. As shown in Supplementary Figure 12, for our PTCDI||I₂ full cell, a voltage drop of 0.4 V needs 78.2 s; in contrast, for the Zn||I₂ full cell, the time required for the same voltage drop is as less as 27.6 s. Clearly, the PTCDI||I₂@AC full cell has a much lower self-discharge rate than that of the conventional Zn||I₂@AC full cell.

The results have been added in Supplementary Figure 12 and described/discussed accordingly in the revised manuscript (Page 9-10, Line 215-228) as follows:

“Originated from the dissolubility of iodine and iodine species in the aqueous environment of the batteries, self-discharge behavior is common for the aqueous iodine-cathode battery systems.³⁻⁶ How to reduce the self-discharge rate effectively has been an intriguing but challenging issue. In the cases of the conventional aqueous metal||I₂ battery systems, iodine anionic species can diffuse into the vicinity of the metal anode and induce the formation of electrochemically inactive complexes, resulting in the irreversible loss of iodine elements and severe self-discharge behaviour.³⁻⁶ Encouragingly, the anode material adopted in this work was PTCDI, an organic compound featuring intrinsic inertness to various iodine anionic species. Iodine anionic species were unable to react with the PTCDI anode to form electrochemically inactive complexes. As a result, the PTCDI||I₂ full cell in this work displayed a much lower self-discharge rate than that of the conventional Zn||I₂ full cell (Supplementary Figure 12). Specifically, the time required for a voltage drop of 0.4 V extended from 27.6 s for the latter to 78.2 s. That is, the self-discharge rate of the PTCDI||I₂ full cell was reduced to 35.3% as that of the Zn||I₂ full cell.”

Supplementary Figure 12. Comparison of the self-discharge rate between the PTCDI||I₂@AC full cell and Zn||I₂@AC full cell.

Reviewer #2 (Remarks to the Author):

Comments:

This paper describes a very interesting concept of new type of electrochemical cell based on cheap materials. The authors were able to come up with explanations and simple models to analyse and understand the measured data. In that respect they used a number of relevant analytical tools to proof the behaviour and existence of the electro-active species.

Answer: We highly appreciate the reviewer for raising the constructive comments and positive remarks on our manuscript. Following this reviewer's suggestions, we have reorganized the figures/main text and added new results/discussion based on the additional experiments in the revised manuscript and SI. Please refer to our point-by-point response below.

Unfortunately, the paper is quite chaotic, and actually it is way too difficult to read – that is why it took me a little longer to go through. Figures are complicated with often irrelevant information, whereas important figures are placed in the supporting information, where they do not belong. The paper also lacks of a simple overall schematic of the cascade cell from where they authors can start to explain their work and results – this would absolutely help. In a similar aspect, why mentioning the important electrochemical steps for the first time in the captions of Figure 4, and then only at line 314 – this could for instance govern the schematic figure in the beginning. On the other hand, Fig. 3 is a figure where we can indeed read relevant data, important for the understanding of the story, this is how it should look like.

Answer: We are very grateful to the reviewer for providing all these important advices. Following this reviewer's suggestion, we have reorganized the figures and main text in the revised manuscript and SI as follows:

- i) We have removed the original panels of Figure 4 about the comparison of the operating voltage range and the pouch cell from the manuscript and placed them in SI as Supplementary Figure 22 and 24, respectively.
- ii) The schematic illustration of the symmetric cell originally located in SI has been placed in the revised manuscript as Figure 4d and Figure 4e.
- iii) The contribution ratio of the capacitive effect of the cascade cell has been placed in the revised manuscript as Figure 4j.
- iv) We have added a simple overall schematic illustration of the cascade cell as Figure 1d.
- v) The electrochemical steps of the PTCDI||I₂ single cell and cascade cell have been added to Page 5 (Line 123-127) and Page 6 (Line 145-157) in the revised manuscript as follows:

Main text: *“In addition, considering the different reaction potentials between I⁻/I⁰ (0.53 V vs. SHE) and I⁰/I⁺ (1.07 V vs. SHE),³ the I⁻/I⁰||I⁰/I⁺ symmetric cell was self-constructed in the aqueous KCl + I₂(aq) mixed electrolyte, which corresponds to the first electrochemical step in the cascade cell and will be discussed later.”*

Main text: *“As discussed above, in addition to working stably in aqueous saturated KCl electrolyte as a single cell [Equations (1-2)], the PTCDI||I₂ full cell can also work stably as a cascade cell in aqueous saturated KCl + I₂ (aq) mixed electrolyte, which consists of an I⁻/I⁰||I⁰/I⁺ symmetric cell [Equations (3-4)] and a PTCDI||I₂@AC single cell [Equations (5-6)] (Figure 1d). According to the different components of the electrolyte, the details of the PTCDI||I₂ single cell in a saturated KCl electrolyte [Equations (1-2)] and a PTCDI||I₂ cascade cell in a mixed electrolyte [Equations (3-6)] will be discussed later.*

With respect to the data evaluation, I have to following comments:

--- Regarding the current evaluation in the CV curves, wouldn't it be better to have deconvolution of the various peaks as the peaks are quite broad and processes are overlapping. In the same respect, in Fig s3b, peak 4 may consist of two peaks, similarly in Fig s7b, where peak 3 obviously has composed of at least two peaks. Besides, the width of the scan rates is small, i.e. only a factor of 2 and 4 respectively – it is questionable whether this is enough to use the underlying model to conclude on the values given.

Answer: We are very grateful to the reviewer for raising these constructive comments. Following this reviewer's suggestion, we first carried out additional measurements on the CV curve of the PTCDI electrode at various scan rates ranging from 5 mV s⁻¹ to 45 mV s⁻¹ in a three-electrode system (Supplementary Figure 3c). To determine how many peak(s) involved in Peak 4 (located at -0.8 V to -0.3 V), we have measured additional CV curve of the PTCDI electrode in a three-electrode system at a small scan rate of 0.2 mV s⁻¹ (Supplementary Figure 3d). In addition, we have analyzed the kinetic behaviors of the PTCDI electrode. The new results have been added (Supplementary Figure 3c-f) and described in the revised SI (Page 5-6, Line 68-96) as follows:

"To further elucidate the kinetic behaviours, CV tests were performed at various scan rates from 5 mV s⁻¹ to 45 mV s⁻¹, which was a large enough range to evaluate the kinetic behaviour.¹⁰⁻¹³ With increasing scan rates, the intensity of the redox peaks increased gradually, and the CV curves retained similar shapes, confirming the excellent high rate tolerance of the PTCDI electrode (Supplementary Figure 3c).¹⁴ To determine whether the relatively broad peak (named peak 4) within -0.8 V to -0.3 V consists of one peak or not, the PTCDI electrode was tested in a three-electrode system at an ultralow scan rate of 0.2 mV s⁻¹. As presented in Supplementary Figure 3d, it is clear that peak 4 consists of one peak. The broadening of the peaks at high scan rates should be mainly attributed to the increased polarization.^{2, 10-13}

In principle, the relationship between the peak current (i_p) and scan rate (v) can be expressed by the power law:¹⁵⁻¹⁷

$$i = av^b \quad (1)$$

where i and v represent the generated current and applied scan rate, respectively, and a and b are constants. In detail, $b \sim 0.5$ indicates a diffusion-controlled process, while $b \sim 1$ indicates a capacitance-dominated process.¹⁵⁻¹⁷ By plotting $\lg i_p$ vs. $\lg v$ (Supplementary Figure 3e), the b values of the five marked redox peaks are calculated to be 0.60 (peak 1), 0.79 (peak 2), 0.80 (peak 3), 0.80 (peak 4), and 0.54 (peak 5), manifesting the combination of diffusion-controlled and capacitance-dominated behaviours of the whole redox process.¹⁸

For further determination of the capacitive contribution, the current density (i) at a fixed potential can be divided

into the capacitance-dominated contribution (k_1v) and the diffusion effect ($k_2v^{1/2}$), and the relationship can be expressed as follows:¹⁵⁻¹⁷

$$i = k_1v + k_2v^{1/2} \quad (2)$$

As a typical example, at a scan rate of 15 mV s^{-1} , the capacitance-controlled contribution approached 69.0%. Moreover, the proportion of capacitive-domination increased with increasing scan rate and reached 83.9% at 45 mV s^{-1} (Supplementary Figure 3f). The high capacitive contribution of the PTCDE electrode is a common phenomenon and has been widely reported in previous works.^{1, 3, 4, 8, 19-22}

Supplementary Figure 3. *c*, CV curves of the PTCDE electrode at various scan rates. *d*, Peak 4 in the CV curve of the PTCDE electrode in the three-electrode system at 0.2 mV s^{-1} . *e*, The corresponding $\lg i_p$ vs. $\lg v$ plots at different redox peaks. *f*, Contribution ratio of the capacitive- and diffusion-controlled process at different scan rates.

Moreover, following this reviewer's suggestion, we have also carried out additional CV measurements for the PTCDE||I₂ full cell in saturated KCl electrolyte at various scan rates ranging from 2 mV s^{-1} to 30 mV s^{-1} (Supplementary Figure 8d). To determine whether the relatively broad peak (Peak 3) located at 1.8 V to 2.1 V consists of one peak or not, we have measured the CV curve of the full cell at a small scan rate of 1 mV s^{-1} (Supplementary Figure 8e). Next, we have also analyzed the kinetic behaviors of the PTCDE||I₂ full cell (Supplementary Figure 8f–g). The new results have been added in Supplementary Figure 8 and addressed in the revised SI (Page 12-13, Line 174-187) and manuscript (Page 8, Line 182-183) as follows:

SI: "Various scan rates from 2 mV s^{-1} to 30 mV s^{-1} for the PTCDE||I₂ full cell in saturated KCl electrolyte were tested, and the width of the scan rates was large enough to evaluate the kinetic behaviour.¹⁰⁻¹³ With increasing scan rate, the shape of the CV curves was well maintained (Supplementary Figure 8d). To determine whether the relatively broad peak (named peak 3) within 1.8 V to 2.1 V consists of one peak or not, the full cell was tested at a small scan rate of 1 mV s^{-1} . As presented in Supplementary Figure 8e, peak 3 consists of one peak. The broadening of the peaks at high scan rates should be mainly attributed to the increased polarization.^{2, 10-13} Moreover, the *b* values of

the redox peaks were calculated to be 0.64 (peak 1), 0.52 (peak 2), 0.78 (peak 3), 0.64 (peak 4), and 0.68 (peak 5) according to equation (1), suggesting the combined mechanism of the whole redox process (Supplementary Figure 8f). As a typical example, at a scan rate of 10 mV s^{-1} , the capacitance-controlled contribution approached 53.3%. Moreover, it increased with increasing scan rate and reached 72.1% at 30 mV s^{-1} (Supplementary Figure 8g), implying the outstanding rate performance of the full cell.^{23, 24}

Main Text: “In addition, the high proportion of capacitive-dominated effects at high scan rates implies the outstanding rate performance of the full cell (Supplementary Figure 8d-g).^{44, 45}”

Supplementary Figure 8. **d**, CV curves of the full cell at various scan rates. **e**, Peak 3 in the CV curve of the PTCDI||I₂ full cell in a saturated KCl electrolyte at 1 mV s^{-1} . **f**, The corresponding $\lg i_p$ vs. $\lg v$ plots at different redox peaks. **g**, Contribution ratio of the capacitance- and diffusion-controlled process at different scan rates.

Other points:

---In line 122 of the supporting information, you gave a reaction with K^+ , but the product formed has also a proton accepted (middle product) – was wondering whether this is a typo or are protons indeed involved in this reaction?

Answer: We thank the reviewer for pointing out this typo. We have revised the schematics of the redox mechanism for reversible K^+ storage reaction of the PTCDI electrode in Supplementary Figure 5, and interpreted that the redox mechanism of the PTCDI electrode in the revised SI (Page 8, Line 117-119) as follows:

“By combining the aforementioned CV results and actual capacity delivered, the redox mechanism of the PTCDI electrode can be expressed as a stepwise enolization reaction in terms of reversible stepwise intercalation of K^+ .^{1, 4,}

8”

Supplementary Figure 5. Schematics of the redox mechanism for the reversible K^+ storage reaction of the PTCDI electrode.

Moreover, we have added the CV curves of the PTCDI electrode measured in a dilute HCl electrolyte ($\text{pH} \approx 5$) in Supplementary Figure 3b and addressed accordingly in the revised SI (Page 5, Line 65-67) as follows:

“As presented in Supplementary Figure 3b, no redox peaks were observed, verifying that protons had no effect on the charge/discharge process of the PTCDI electrode.”

Supplementary Figure 3. b, CV curves of the PTCDI electrode through a three-electrode system in dilute HCl solution ($\text{pH} \approx 5$) at 10 mV s^{-1} .

--- In Fig 2f, a cycle number of 90k is given meaning 90,000. This seems to be a lot: hence considering a 10C charge and discharge, a full cycle takes 12 minutes 0.2 hours. Every year has 8760 hours, which means at a charge/discharge of 10C it 90,000 cycles will take just more than 2 years. Hence, was wondering whether this is indeed the case – or at least provide the C-rate somewhere.

Answer: We thank the reviewer for raising this concern. As the theoretical specific capacity of I_2 ($\text{I}^-/\text{I}^0/\text{I}^+$) is 422 mAh g^{-1} , 1 C for I_2 is 422 mA g^{-1} (*Energy Environ. Sci.* 2021, **14**, 407-413; *Nat. Commun.* 2021, **12**, 170; *Angew. Chem. Int. Ed.* 2022, **61**, e202113576). Regarding the results in Figure 2f, the current density applied was 40 A g^{-1} , corresponding to 94.8 C. Consequently, it took 1890 hours, i.e. 79 days, in total to complete the 90000 cycles. Following this reviewer’s suggestion, we have added the corresponding C-rate value together with the current density in the revised manuscript (Page 4, Line 86-87; Page 9, Line 197-198) as follows:

“an ultralong lifespan (92000 cycles at 40 A g^{-1} (94.8 C) and an extremely high rate tolerance (104 mAh g^{-1} at 160 A g^{-1}),”

“Remarkably, a discharge capacity of 154 mAh g^{-1} was achieved after 92000 cycles at 40 A g^{-1} (94.8 C) (Figure 2f).”

--- In summary: Please reorganise the paper so that the readers are being triggered, and not to have figuring out themselves where the relevant information has been placed. This too, includes selecting the important figures – which one to put in the body of the paper, and which in the supporting information. With respect to data analysis,

reanalysing and deconvolution of the CV curves would be essential to better calculate the fractions of the capacitive-controlled vs the diffusive-controlled behaviour.

Answer: Once again, we highly appreciate the reviewer for raising all the constructive comments. Following this reviewer's suggestion, we have reorganized the figures/main text and added new results for the CV curves based on additional experiments together with the analysis of the kinetic behaviors for the PTCDI electrode and the PTCDI||I₂ full cell in the revised manuscript and SI. Thanks to the re-organization and revisions, the readability and quality of this work have been improved substantially.

Reviewer #3 (Remarks to the Author):

Comments:

The cell voltage of the proposed organic cathode||I₂ cascade battery is 2.5 V, which is well above the thermodynamic potential of the water splitting. The authors provided the LSV data to support the there is no water splitting. But it is not convincing.

Answer: We thank the reviewer for raising this concern. First, the LSV measurement has been widely used to evaluate the electrochemical potential window of the electrolyte (*Nat. Energy* 2019, **4**, 495-503; *Angew. Chem. Int. Ed.* 2021, **60**, 7366-7375; *Science* 2015, **350**, 938-943; *Nat. Mater.* 2020, **19**, 1006-1011). In our work, the electrochemical potential window of the aqueous electrolyte was determined by using the three-electrode system (counter electrode: Pt electrode; reference electrode: standard Ag/AgCl electrode) with the Ketjen Black electrode (Ketjen Black:PVDF = 9:1) as working electrode (*Angew. Chem. Int. Ed.* **2016**, **55**, 12768-12772). Following this reviewer's suggestion, we have added the results in Supplementary Figure 14 and addressed accordingly in the revised SI (Page 20, Line 265-269) and manuscript (Page 15, Line 312-315) as follows:

SI: "As shown in Supplementary Figure 14, the slight bulge within the potential range of 0.6 V to 0.9 V corresponds to the oxidation reaction of I⁰/I⁺ in the mixed electrolyte,^{13, 31} which is an indispensable part of the I⁻/I⁰||I⁰/I⁺ symmetric cell. Hence, the electrochemical potential window of the mixed electrolyte was determined to be 2.60 V, which is large enough to guarantee the stable work of the cascade cell."

Main text: "The operational voltage window of the saturated KCl + I₂(aq) mixed electrolyte was 2.60 V, slightly broader than that of the pure saturated KCl electrolyte, which may be ascribed to the lower amount of free water when additional I₂ molecules were dissolved in the electrolyte (Supplementary Figure 14).⁶⁴"

As well-demonstrated in the earlier works (*Nat. Mater.* 2018, **17**, 543-549; *Angew. Chem. Int. Ed.* 2020, **59**, 9377-9381; *Adv. Funct. Mater.* 2019, **29**, 1902653), highly concentrated electrolytes can be used to broaden the electrochemically-stable potential window by decreasing the number of free water molecules in the electrolyte, and the water splitting can be significantly suppressed in this case. Apart from the saturated KCl, owing to the solubility of I₂ in free water, the introduction of I₂ was of benefit to diminish activity of free water as a result of the reduced content of free water (*Nano Lett.* 2015, **15**, 5982-5987; *ACS Energy Lett.* 2017, **2**, 2674-2680). For this reason, the cascade cell can work steady in the mixed electrolyte.

Following this reviewer's suggestion, to further ascertain that there is no water splitting, we have monitored *in-situ* gas pressure variation during the CV measurements of the cascade cell at the voltage ≤ 2.5 V. The results have been added in Supplementary Figure 19 and addressed accordingly in the revised manuscript (Page 17, Line 379-382; Page 23, Line 511-513) as follows:

"In addition, no fluctuation in gas pressure was detected during the whole process of the cascade cell, indicating that no H_2 or O_2 evolution reaction occurred (Supplementary Figure 19).⁶⁴ These results consolidate the electrochemical stability of the mixed electrolyte within 2.5 V."

"An in-situ pressure test during the CV test was performed to determine the H_2 and O_2 evolution through the pressure transducer (CYYZ11, Starsensor, China)."

Supplementary Figure 14. LSV curve of the saturated KCl + $I_2(aq)$ mixed electrolyte.

Supplementary Figure 19. In situ pressure test during the CV measurement of the PTCDI || I_2 cascade cell in mixed electrolyte at 10 mV s^{-1} for 10 cycles.

--- There is no mention of percentage of specific capacity retention against the theoretical specific capacity of the cell. Figure 2F, no explanation for linear decrease in specific capacity with cycle number.

Answer: We thank the reviewer for raising the valuable point. Following this reviewer's suggestion, we have added the percentage of specific capacity retention against the theoretical specific capacity of the cell in the revised manuscript (Page 9, Line 198-201) as follows:

"The percentage of specific capacity retention against the theoretical (422 mAh g^{-1}) and initial (363.4 mAh g^{-1}) specific capacity of the cell after 92000 cycles were calculated to be 36.5% and 42.4%, respectively, delivering outstanding decay ratios of 0.7% and 0.6% per thousand cycles, respectively."

The linear decrease of the specific capacity with the cycle number is mainly attributed to the solubility of iodine molecules and iodine anionic species derived from the I₂@AC cathode, which is a common phenomenon for the aqueous iodine-cathode battery systems (*Energy Environ. Sci.* 2021, **14**, 407-413; *Nat. Commun.* 2021, **12**, 170; *Adv. Mater.* 2021, **33**, 2006897). Fortunately, the PTCDI compound used in this work possesses a large π -conjugated structure with strong π - π interactions and intermolecular hydrogen bonding, which can endow the anode excellent structural stability during the repeated charge/discharge process (*Nat. Sustain.* 2022, **5**, 225-234; *Nat. Energy* 2019, **4**, 495-503; *Chem. Eur. J.* 2020, **26**, 17559-17566). Benefited from the intrinsic inertness to various iodine anionic species of the PTCDI anode, ~154 mAh g⁻¹ was still delivered at 40 A g⁻¹ after 92000 cycles, far exceeding other reported aqueous iodine-cathode batteries (*Nat. Commun.* 2021, **12**, 170; *Adv. Mater.* 2020, **32**, 2004240; *Adv. Mater.* 2021, **33**, 2006897; *Energy Environ. Sci.* 2021, **14**, 407-413; *Angew. Chem. Int. Ed.* 2021, **60**, 3791-3798). Following this reviewer's suggestion, we have addressed this point in the revised manuscript (Page 9, Line 202-205) as follows:

".....and the decrease in the specific capacity with cycle number were mainly attributed to the solubility of iodine molecules and iodine anionic species derived from the I₂@AC cathode, which is a common phenomenon in aqueous iodine-cathode battery systems.^{3, 5, 6, 13, 14"}

--- *The oxidative stability of PTCDI is questionable (<https://doi.org/10.1002/chem.202003624>).*

Answer: We thank the reviewer for raising this concern. In fact, in the reference mentioned by the reviewer (<https://doi.org/10.1002/chem.202003624>), on the contrary to raising the concern about the oxidative stability of PTCDI, the authors confirmed the long-term cycling stability of PTCDI at high oxidation potential: *"Organic cathode materials are known for their poor cycle life performance, due to their structural instability particularly at high oxidation potentials and their tendency to dissolve in organic battery electrolytes. In contrast, the long-term cycling stability of PTCDI is exceptionally enhanced, possibly due to the larger π -conjugated structure and therefore increased π - π interactions and strong intermolecular H-bonding of PTCDI molecules."* Besides the structural stability (*Chem. Eur. J.* 2020, **26**, 17559-17566), PTCDI is insoluble in water and capable to work stably as anode in aqueous high-voltage potassium-ion full batteries at the voltage range of 0-2.6 V (*Nat. Energy* 2019, **4**, 495-503) and 0-2.5 V (*Nat. Sustain.* 2022, **5**, 225-234). Following this reviewer's suggestion, we have emphasized this point in the revised manuscript (Page 9, Line 207-210) as follows:

"Such an ultralong lifespan with high capacity at a high rate was attributed to the inertness to iodine anions and the layered structure of the PTCDI anode with large interplanar spacing, as well as the stability of the PTCDI anode at high oxidation potentials owing to the large π -conjugated structure.^{33, 48, 49"}

--- *There was no formation of I₃⁻, as supported from the UV-visible spectra, XPS, Raman and FTIR. What could be the possible reason?*

Answer: We thank the reviewer for raising this question. The absence of I₃⁻ is mainly attributed to the presence of K⁺ ions in aqueous electrolyte, which is beneficial to a direct conversion between I⁻ and I⁰. According to our DFT

calculations, the lowest value of Gibbs free energy change (ΔG) was observed when the Γ^-/Γ^0 conversion chemistry occurred in the K^+ environment (Figure 1a). In this way, the aqueous electrolyte containing K^+ holds the maximum potential to accelerate the Γ^-/Γ^0 conversion, superior to other cations (*Energy Environ. Sci.* 2021, **14**, 407-413; *Angew. Chem. Int. Ed.* 2021, **60**, 3791-3798). In the present work, saturated KCl solution was utilized as electrolyte for the PTCDI|| I_2 full cell, which facilitated the fast reaction rate of the Γ^-/Γ^0 conversion. Following this reviewer's suggestion, we have carried out additional CV measurements of the I_2 @AC electrode in a saturated KCl electrolyte and added the results in Supplementary Figure 13 and addressed them in the revised SI (Page 19, Line 255-259) and manuscript (Page 12, Line 266-269) as follows:

SI: "As shown in Supplementary Figure 13, the CV of the I_2 @AC electrode in a saturated KCl electrolyte was tested in a three-electrode system. Only a pair of redox peaks corresponding to Γ^-/Γ^0 was observed, and no other redox peaks corresponding to Γ^-/I_3^- were detected, confirming a direct conversion between Γ^- and Γ^0 .^{13, 31, 40} These results are consistent with the DFT result in Figure 1a.^{13, 25, 30, 31, 41}"

Main text: "No characteristic peak of I_3^- (290 nm) was observed, which may mainly be attributed to the presence of K^+ ions in the aqueous electrolyte, which was beneficial to the direct conversion between Γ^- and Γ^0 and consistent with the DFT results in Figure 1a (Supplementary Figure 13).^{2, 3, 5}"

Supplementary Figure 13. CV curve of the I_2 @AC electrode in a three-electrode system (working electrode: I_2 @AC; counter electrode: platinum; reference electrode: standard Ag/AgCl; electrolyte: saturated KCl solution) at 2 mV s^{-1} .

--- Line no. 185-189, the explanation is repeated.

Answer: We thank the reviewer and have deleted the repeated section. Please see the details in the revised manuscript (Page 10, Line 241-242) as follows:

".....showing excellent promise for practical applications. The obtained results promote the electrochemical performance of aqueous I_2"

--- Can author provides the details about specific capacity calculations?

Answer: We thank the reviewer for raising this constructive comment. Following the reviewer's suggestion, we have added the detailed method for the specific capacity calculations in the revised manuscript (Page 23, Line 513-517) as follows:

"The specific capacity C (mAh g^{-1}) was calculated by the following equation:³²

$$C = I\Delta t / 3.6m$$

where I (A) is the applied current, Δt (s) is the corresponding charge or discharge time, and m (g) is the weight of I_2 in the $\text{I}_2@\text{AC}$ electrode."

--- The quantitative adsorption of I_2 on activated carbon is not mentioned.

Answer: We thank the reviewer for raising this valuable point. Following this reviewer's suggestion, we have calculated the content of I_2 by thermogravimetric analysis (TGA). The results have been added in Supplementary Figure 7b and addressed accordingly in the revised SI (Page 10, Line 147-149) and manuscript (Page 6, Line 143-145; Page 22, Line 495-497) as follows:

SI: "As presented in Supplementary Figure 7b, the second stage corresponds to the mass loss of I_2 , and the content of I_2 in the $\text{I}_2@\text{AC}$ was determined to be 47.2%."

Main text: " $\text{I}_2@\text{AC}$ (the content of I_2 was 47.2%) synthesized through a facile physical adsorption method was utilized as the cathode (Supplementary Figure 7)."

Main Text: "Thermogravimetric analysis (TGA) was performed to determine the content of iodine with a TG-DSC analyser (NETZSCH, STA 449 F3, Germany), and the heating rate was $5\text{ }^\circ\text{C min}^{-1}$ from room temperature to $500\text{ }^\circ\text{C}$ under an Ar atmosphere."

Supplementary Figure 7. b, TGA curve of the synthesized $\text{I}_2@\text{AC}$ powder.

--- Can author provide the kinetic details of the cascade reactions?

Answer: We are very grateful to the reviewer for providing this constructive comment. Following this reviewer's suggestion, we have measured the CV curves of our cascade cell at various scan rates ranging from 3 mV s^{-1} to 15 mV s^{-1} in the mixed electrolyte. According to the earlier works (*Nat. Commun.* 2022, **13**, 125; *Joule* 2021, **5**, 2993-3005), such a width of the scan rates is big enough to evaluate the kinetic behavior. The results have been added in Figure 4j and Supplementary Figure 18 and addressed accordingly in the revised manuscript (Page 17, Line 375-379) and SI (Page 5-6, Line 78-83; Page 6, Line 88-91; Page 24, Line 292-299) as follows:

Main text: "The kinetic behaviours of the cascade cell were evaluated under various scan rates, and the whole

redox process was regulated by the combination of diffusion-controlled and capacitance-dominated effects (Supplementary Figure 18). As shown in Figure 4j, the cascade cell displays a high capacitive contribution under high scan rates, implying the outstanding rate performance of the cell.”

SI: “In principle, the relationship between the peak current (i_p) and scan rate (v) can be expressed by the power law:¹⁵⁻¹⁷

$$i = av^b \quad (1)$$

where i and v represent the generated current and applied scan rate, respectively, and a and b are constants. In detail, $b \sim 0.5$ indicates a diffusion-controlled process, while $b \sim 1$ indicates a capacitance-dominated process.¹⁵⁻¹⁷”

SI: “For further determination of the capacitive contribution, the current density (i) at a fixed potential can be divided into the capacitance-dominated contribution (k_1v) and the diffusion effect ($k_2v^{1/2}$), and the relationship can be expressed as follows:¹⁵⁻¹⁷

$$i = k_1v + k_2v^{1/2} \quad (2)$$

SI: “Various scan rates from 3 mV s^{-1} to 15 mV s^{-1} for the PTCDI|| I_2 cascade cell were used to evaluate the kinetic behaviour.^{11, 12} As presented in Supplementary Figure 18a, the CV curves displayed similar shapes, confirming the excellent high rate tolerance of the cascade cell.¹⁴ According to equation (1), the b values of the seven marked redox peaks were calculated to be 0.24 (peak 1), 0.76 (peak 2), 0.46 (peak 3), 0.88 (peak 4), 0.69 (peak 5), 0.73 (peak 6), and 0.27 (peak 7), indicating the combination of diffusion-controlled and capacitance-dominated behaviours of the whole redox process (Supplementary Figure 18b).¹⁸”

Supplementary Figure 18. **a**, CV curves of the PTCDI|| I_2 cascade cell in mixed electrolyte at various scan rates. **b**, The corresponding $\lg i_p$ vs. $\lg v$ plots at different redox peaks.

Figure 4. j, Contribution ratio of the capacitance- and diffusion-controlled process of the PTCDI || I₂ cascade cell at different scan rates.

--- Figure 2C and 2D, CV and alpha graph, it is not clear about the practical capacity of the battery. The high capacity was due to the dip discharging. Please explain.

Answer: We thank the reviewer for raising this important comment. To make it more specific, the y-axis has been adjusted from “current” to “specific current” in Figure 2c. The two-electron transfer mode (I⁻/I⁰/I⁺) doubles the conventional theoretical specific capacity (I⁻/I⁰) to 422 mAh g⁻¹ and the high capacity of the full cell originates from the two-electron transfer mode of the I₂@AC cathode (I⁻/I⁰/I⁺) (*Energy Environ. Sci.* 2021, **14**, 407-413; *Nat. Commun.* 2021, **12**, 170; *Angew. Chem. Int. Ed.* 2022, **61**, e202113576). As shown in Figure 2c, the redox peaks appeared within the voltage of 0.3–2.4 V. To make sure that all redox peaks can be completely displayed, we have carried out additional GCD measurements on the full cell in the voltage range of 0.3–2.4 V. The results have been added in Supplementary Figure 9 and described accordingly in the revised SI (Page 14, Line 193-202) and manuscript (Page 8, Line 186-187) as follows:

SI: “To ensure that all redox peaks can be completely displayed, the voltage range of the PTCDI || I₂ full cell was set to 0.3-2.4 V. As shown in Supplementary Figure 9, a discharge capacity of 300 mAh g⁻¹ was delivered, which still far exceeded that of all reported aqueous iodine-cathode batteries at such a high current density.^{13, 25, 29-34} An additional discharge capacity of 24 mAh g⁻¹ was delivered when the cut-off voltage of the full cell was set to 0, which only accounted for 7.4% of the total capacity (324 mAh g⁻¹ in Figure 2d). Therefore, the cut-off voltage of the full cell was not the main reason for the high capacity delivered. Instead, a slightly higher capacity output was obtained if the cut-off voltage was set to 0, and the same method has also been reported in previous works.^{2, 8, 22}”

Main text: “The high capacity mainly originated from the two-electron transfer reaction of the I₂ cathode and was not greatly affected by the cut-off voltage of the cell (Supplementary Figure 9).”

Figure 2. c Typical CV curve of the PTCDI || I₂ full cell in a saturated KCl electrolyte.

Supplementary Figure 9. The typical GCD curve of the PTCDI||I₂ full cell in a saturated KCl electrolyte within the voltage range of 0.3-2.4 V.

--- Figure S3 D, shows the capacitive contribution of the electrode is very high compare to the diffusion controlled. Could you please explain it's contribution to specific capacity of the battery.

Answer: We thank the reviewer for raising the constructive comment. Following this reviewer's suggestion, we have added more explanation on the capacitive- and diffusion-controlled contribution to the specific capacity of the battery in the revised SI (Page 6, Line 92-102) as follows:

"As a typical example, at a scan rate of 15 mV s⁻¹, the capacitance-controlled contribution approached 69.0%. Moreover, the proportion of capacitive-domination increased with increasing scan rate and reached 83.9% at 45 mV s⁻¹ (Supplementary Figure 3f). The high capacitive contribution of the PTCD electrode is a common phenomenon and has been widely reported in previous works.^{1, 3, 4, 8, 19-22} The high capacitive contribution means that the kinetic behaviour of the electrode is mainly dominated by the capacitive effect, and the transport of K⁺ ions is not the rate-limiting factor.²¹ In other words, the transport of K⁺ ions is fast enough under high operational current densities. In addition, the capacitive effect renders more charge transfer than volume lattice diffusion and thus can help to retain the capacity at high operational current densities.^{23, 24}"

--- Figure S7, shows the shift in peak potential of the full cell voltammogram of PTCDI||I₂ in KCl solution. Also it is showing two cathodic peaks after 40th and 90th scan. The explanation provided in line no. 149 is insufficient.

Answer: We thank the reviewer for raising this important point. The cathodic peaks corresponded to the step-wise de-intercalation of K⁺ ions from the PTCDI electrode and the reduction process of the I₂ electrode. The cathodic peaks after the 2nd scan in original Figure S7a (Supplementary Figure 8a in the revised SI) were determined based on the corresponding partial enlargement of the peaks. The results have been added in Supplementary Figure 8b and described accordingly in the revised SI (Page 12, line 160-164) as follows:

"As presented in Supplementary Figure 8a, there are two cathodic peaks below 1.2 V after the 2nd scan, but the lower potential position of the cathodic peaks overlapped with the 40th scan curve. For clarity, the cathodic peaks below 1.2 V after the 2nd scan was enlarged (Supplementary Figure 8b). It is clear that there are two cathodic peaks below 1.2 V after the 2nd scan. The cathodic peaks corresponded to the step-wise de-intercalation of K⁺ ions from the PTCDI electrode and the reduction process of the I₂ electrode."

Moreover, following this reviewer's suggestion, we have carried out additional electrochemical impedance spectroscopy (EIS) measurements to investigate the shift in peak potential of the full cell voltammogram of PTCDI||I₂ in KCl solution. The results have been added in Supplementary Figure 8c and addressed accordingly in the revised SI (Page 12, Line 164-173) and manuscript (Page 8, Line 174-176) as follows:

SI: "To investigate the phenomenon of the shift in peak potential of the full cell, electrochemical impedance spectroscopy (EIS) measurements were performed after the 2nd, 40th and 90th scans at 10 mV s⁻¹. As shown in Supplementary Figure 8c, the EIS curves after different scan cycles presented similar shapes and the positions shifted slightly to the left with the increased scan cycles. Notably, the intersection of the EIS curve with the horizontal axis presents the ohmic resistance of the cell.²⁵⁻²⁸ The smaller the intercept was, the smaller the ohmic resistance.²⁵⁻²⁸ To make it clearer, the inset shows the enlarged zone of the curves at high frequency. It is obvious that the ohmic resistance of the cell decreased with increasing scan cycles."

Main text: "With an increase in the cycle number, the initial redox peaks gradually shifted towards the positive direction and finally stabilized due to the activation process, as verified by the reduced ohmic resistance (Supplementary Figure 8a-c).^{26, 43}"

Supplementary Figure 8. **a**, CV curves of the PTCDI||I₂ full cell in a saturated KCl electrolyte at a scan rate of 10 mV s⁻¹. **b**, The corresponding partial enlargement of the 2nd CV curve of the full cell. **c**, EIS curves of the full cell after the 2nd, 40th and 90th CV tests at 10 mV s⁻¹.

--- Figure S10, CV shows substantial increase in ohmic resistance of the PTCDI electrode after 3rd, 4th and 5th scan. Explanation is not provided.

Answer: We thank the reviewer for raising this comment. Following this reviewer's suggestion, we have carried out additional EIS measurements of the PTCDI electrode after different scan cycles to determine the variation in the ohmic resistance. The results have been added in Supplementary Figure 6 and addressed accordingly in the revised SI (Page 9, Line 127-139) as follows:

“As presented in Supplementary Figure 6a, the CV curves of the PTCDI electrode in the saturated KCl + I₂(aq) mixed electrolyte were identical to those in the pure saturated KCl electrolyte (Supplementary Figure 3a), demonstrating the intrinsic characteristics of inertness to various iodine anionic species of the PTCDI electrode. Since the voltage at the beginning of the test was 0.16 V, the curves in the vicinity of 0.2 V between the 1st scan cycle and the sequential scan cycles were nonclosed. To determine the variation in the ohmic resistance of the PTCDI electrode after different scan cycles, corresponding electrochemical impedance spectroscopy (EIS) measurements were performed. As shown in Supplementary Figure 6b, the intersection of the EIS curve with the horizontal axis after different scan cycles effectively overlapped, illustrating that the ohmic resistance of the PTCDI electrode remained stable without significant variation and consolidating that the PTCDI electrode can work stably in the mixed electrolyte.²⁵⁻²⁸”

Supplementary Figure 6. a, CV curves of the PTCDI electrode in a three-electrode system (working electrode: PTCDI; counter electrode: platinum; reference electrode: standard Ag/AgCl; electrolyte: saturated KCl + I₂(aq) mixed solution) at 10 mV s⁻¹. **b,** The corresponding EIS curves.

REVIEWER COMMENTS

Reviewer #1 (Remarks to the Author):

2nd Review report on: Aqueous organic anode//iodine cascable battery with ultra-long lifespan, high energy and power density

The authors have convincingly addressed all my concerns raised. I can therefore now recommend publication of the revised manuscript as it is in Nature Communications.

Reviewer #3 (Remarks to the Author):

The authors have answered the concern raised by reviewers but still some more explanation is necessary

The question raised by the second reviewer regarding the ' - In Fig 2f, a cycle number of 90k is given meaning 90,000. This seems to be a lot: hence considering a 10C charge and discharge, a full cycle takes 12 minutes 0.2 hours. Every year has 8760 hours, which means at a charge/discharge of 10C it 90,000 cycles will take just more than 2 years. Hence, was wondering whether this is indeed the case – or at least provide the C-rate somewhere' is not answered. Moreover, I wonder how author has prevented the evaporation and crystallization of electrolyte in such an extremely long time span.

Regarding the question of high capacity and dip discharge, Figure 2C shows the reduction peak $\sim 0.6V$ and for the complete discharge, the cutoff voltage is assigned to 0.3V. It certainly have the effect on specific capacity and columbic efficiency. But I could not see change in the values of specific capacity and columbic efficiency in the main manuscript. What will be the practical discharge voltage of the cell ???

Dear Reviewers,

We would like to thank you for taking time to evaluate our submission (Manuscript ID: NCOMMS-22-09631B) and raising the positive remarks and constructive comments, which have helped us improved the quality of our manuscript. We have thoroughly addressed all comments/concerns from the reviewers point-by-point with the associated changes listed in this letter and highlighted in yellow in the revised manuscript and supplementary information. We sincerely hope that we have successfully dealt with all the questions so that you would find our revised manuscript publishable in Nature Communications.

Yours faithfully,

Yan Huang

Reviewer #1 (Remarks to the Author):

Comments:

2nd Review report on: Aqueous organic anode//iodine cascable battery with ultra-long lifespan, high energy and power density

The authors have convincingly addressed all my concerns raised. I can therefore now recommend publication of the revised manuscript as it is in Nature Communications.

Answer: We highly appreciate the reviewer for positive remarks on our manuscript.

Reviewer #3 (Remarks to the Author):

Comments:

The authors have answered the concern raised by reviewers but till some more explanation is necessary

The question raised by the second reviewer regarding the ' - In Fig 2f, a cycle number of 90k is given meaning 90,000. This seems to be a lot: hence considering a 10C charge and discharge, a full cycle takes 12 minutes 0.2 hours. Every year has 8760 hours, which means at a charge/discharge of 10C it 90,000 cycles will take just more than 2 years. Hence, was wondering whether this is indeed the case – or at least provide the C-rate somewhere' is not answered. Moreover, I wonder how author has prevented the evaporation and crystallization of electrolyte in such an extremely long time span.

Answer: We thank the reviewer for pointing out this issue. The theoretical specific capacity of I₂

($I^-/I^0/I^+$) is 422 mAh g⁻¹, hence 1 C for I₂ is 0.422 A g⁻¹ (*Nat. Commun.* 2021, **12**, 170; *Angew. Chem. Int. Ed.* 2022, **61**, e202113576; *Energy Environ. Sci.* 2021, **14**, 407-413). With regard to the results in Figure 2f, the current density applied was 40 A g⁻¹, corresponding to 94.8 C, which means that a full cycle took 0.021 hour. Consequently, it took 1890 hours, i.e. 79 days, in total to complete the 90000 cycles. Following the reviewer's suggestion, we have added the corresponding C-rate value together with the current density to the revised manuscript (Page 4, Line 86-87; Page 9, Line 196, 202; Page 10, Line 234-238; Page 13, Figure 2d, 2f and 2h, Line 260, 268; Page 18, Line 387-388, 390, 392, 395-398; Page 21, Figure 4f, 4k and 4l, Line 442-443, 450) and supplementary information (Page 14, Supplementary Figure 9; Page 17, Line 226, 228; Page 26, Supplementary Figure 20, Line 307; Page 27, Line 311).

Moreover, to prevent the evaporation of electrolyte during the long cycle test (over two months), the full cell was hermetically sealed by laboratory paper film and no crystallization of electrolyte was observed. The related information has been added to the revised manuscript (Page 23, Line 491-494) as follows:

Main text: "The full cell was hermetically sealed by laboratory paper film (PM996, Bemis, America) to guarantee that electrodes were immersed into the electrolyte during the whole test and no crystallization of electrolyte was observed."

Regarding the question of high capacity and dip discharge, Figure 2C shows the reduction peak ~0.6V and for the complete discharge, the cutoff voltage is assigned to 0.3V. It certainly have the effect on specific capacity and columbic efficiency. But I could not see change in the values of specific capacity and columbic efficiency in the main manuscript. What will be the practical discharge voltage of the cell ???

Answer: We thank the reviewer for pointing out this issue. Following the reviewer's suggestion, the results of the change in the values of specific capacity and columbic efficiency have been added to the revised main manuscript (Page 8, Line 186-191) as follows:

Main text: "When the cut-off voltage of the cell was set to 0.3 V, a discharge capacity of 300 mAh g⁻¹ with the coulombic efficiency of 82% was delivered (Supplementary Figure 9). Hence, the cut-off voltage of the full cell had the effect on the specific capacity and coulombic efficiency to some extent. Considering that a slightly higher capacity output was obtained, the cut-off voltage of the full cell was set to 0 in this work."

Considering that the discharge curve has multiple discharge platforms, herein, the average voltage of 1.26 V was adopted to evaluate the practical voltage of the full cell. This voltage is far

higher than the required voltage of some practical applications such as potato clock etc. We have revised the manuscript (Page 8, Line 183-186) as follows:

Main text: "The typical galvanostatic charge/discharge (GCD) curve displayed the maximal plateau at 1.90 V and average voltage of 1.26 V during the discharge process, and a discharge capacity of 324 mAh g⁻¹ with the coulombic efficiency of 69% was delivered at 40 A g⁻¹ (94.8 C) (Figure 2d)."

REVIEWERS' COMMENTS

Reviewer #3 (Remarks to the Author):

The manuscript can be accepted in its present form.

Dear Reviewers,

We would like to thank you for taking time to evaluate our submission (Manuscript ID: NCOMMS-22-09631B) and raising the positive remarks and constructive comments, which have helped us improved the quality of our manuscript.

Yours faithfully,

Yan Huang

Reviewer #3 (Remarks to the Author):

Comments:

The manuscript can be accepted in its present form.

Answer: We highly appreciate the reviewer for positive remarks on our manuscript.